# Reinforcement Teaching

**Calarina Muslimani**[*1,2]          *musliman@ualberta.ca*
**Alex Lewandowski**[*1,2]          *lewandowski@ualberta.ca*
**Dale Schuurmans**[1,3,4]          *daes@ualberta.ca*
**Matthew E. Taylor**[1,4]          *matthew.e.taylor@ualberta.ca*
**Jun Luo**[2]          *jun.luo1@huawei.com*

[1] *Department of Computing Science, University of Alberta*
[2] *Noah's Ark Lab, Huawei Technologies Canada Co., Ltd.*
[3] *Google Brain*
[4] *Alberta Machine Intelligence Institute (Amii)*
[*] *Equal Contribution. Work done while interning at Huawei.*

**Reviewed on OpenReview:** *https://openreview.net/forum?id=G2GKiicaJI*

## Abstract

Machine learning algorithms learn to solve a task, but are unable to improve their ability to learn. Meta-learning methods learn about machine learning algorithms and improve them so that they learn more quickly. However, existing meta-learning methods are either hand-crafted to improve one specific component of an algorithm or only work with differentiable algorithms. We develop a unifying meta-learning framework, called *Reinforcement Teaching*, to improve the learning process of *any* algorithm. Under Reinforcement Teaching, a teaching policy is learned, through reinforcement, to improve a student's learning algorithm. To learn an effective teaching policy, we introduce the *parametric-behavior embedder* that learns a representation of the student's learnable parameters from its input/output behavior. We further use *learning progress* to shape the teacher's reward, allowing it to more quickly maximize the student's performance. To demonstrate the generality of Reinforcement Teaching, we conduct experiments in which a teacher learns to significantly improve both reinforcement and supervised learning algorithms. Reinforcement Teaching outperforms previous work using heuristic reward functions and state representations, as well as other parameter representations.

## 1 Introduction

As machine learning becomes ubiquitous, there is a growing need for algorithms that generalize better, learn more quickly, and require less data. Meta-learning is one way to improve a machine learning algorithm, without hand-engineering the underlying algorithm. Meta-learning is often thought of as "learning to learn" in which the goal is to learn about and improve another machine learning process (Schmidhuber, 1994; Thrun & Pratt, 1998; Hospedales et al., 2022). A variety of sub-domains have emerged that design hand-crafted solutions for learning about and improving a specific component of a machine learning process. The work in these sub-domains focus on solving one specific problem, whether that be finding the best way to augment data (Cubuk et al., 2019), sample minibatches (Fan et al., 2018), adapt objectives (Wu et al., 2018a), or poison rewards (Zhang et al., 2020). Consequently, the meta-learning methods used in these domains are handcrafted to solve the problem and cannot be applied to solve new problems in a different domain.

Current literature fails to recognize that a more general framework can be used to simultaneously address multiple problems across these varied sub-domains. Therefore, this work takes an important step toward answering the following question:

*Can we develop a unifying framework for improving machine learning algorithms that can be applied across sub-domains and learning problems?*

As a crucial step towards this unifying framework, we introduce *Reinforcement Teaching*: an approach that frames meta-learning in terms of learning in a Markov decision process (MDP). Although the individual components of our system are based on previously proposed principles and methods from teacher-student reinforcement learning (Almeida et al., 2021; Garcia & Thomas, 2019; Huang et al., 2019; Cubuk et al., 2019; Ruiz et al., 2019; Campero et al., 2020; Florensa et al., 2018; Fan et al., 2018; Narvekar et al., 2017; Narvekar & Stone, 2019; Zhang et al., 2020; Zhu et al., 2019; Zoph & Le, 2017; Jomaa et al., 2019; Biedenkapp et al., 2020; Sabbioni et al., 2020), learned parameter representations (Harb et al., 2020; Parker-Holder et al., 2020), and learning progress (Oudeyer et al., 2007), our system combines these components into a novel framework that unifies meta-learning approaches and can improve machine learning algorithms across sub-domains. To the best of our knowledge, Reinforcement Teaching is the first attempt at using reinforcement learning as a general-purpose meta-learning solution method.

In Reinforcement Teaching, a teacher learns a policy via reinforcement learning (RL) to improve the learning process of a student. The teacher observes a problem-agnostic representation of the student's behavior and takes actions that adjust components of the student's learning process that the student is unable to change, such as the student's objective, optimizer, data, or environment. The teacher's reward is then based on the student's performance. The choice of action space for the teacher induces different meta-learning problem instances. This allows our single teaching-architecture to learn a variety of policies, such as a curriculum policy to sequence tasks for an RL student or a step-size adaptation policy for a supervised learning student.

Our Reinforcement Teaching framework has several advantages over both gradient-based meta-learning and other RL teaching methods. Like gradient-based meta-learning, our MDP formalism is problem-agnostic and thus does not rely on problem-specific heuristics used in other RL teaching methods (Huang et al., 2019; Almeida et al., 2021; Garcia & Thomas, 2019; Cubuk et al., 2019; Ruiz et al., 2019; Fan et al., 2018; Jomaa et al., 2019). These prior RL teaching methods use hand-designed features for the RL teaching policy. This choice limits the generality of these approaches to the base problems they were intended to solve. Other works use parameter-based state representations which have been shown to learn successful teaching policies for tabular and linear students (Biedenkapp et al., 2020; Zhang et al., 2020; Narvekar et al., 2017; Sabbioni et al., 2020). Although the parameter state representation is problem-agnostic, it is difficult to scale to more complex problems such as if the student uses a deep non-linear neural network. This motivated the development of the parametric-behavior embedder, a problem-agnostic state representation that can be used across different problem settings and can scale to deep non-linear students.

Moreover, although successful, gradient-based meta-learning methods (Finn et al., 2017; Xu et al., 2018; Javed & White, 2019) do not learn a teaching policy and are therefore unable to adapt to the student's needs at each step in the student's learning process. Another limitation of gradient-based meta-learning methods is the requirement that all learning components are fully-differentiable, which is not always possible. One example of a component that is not differentiable is the configuration of an environment, which a teacher policy may control to induce a curriculum for a student.

This paper makes the following contributions:

1. The Reinforcement Teaching framework is formalized as an MDP in which the teacher learns a policy that adapts the student's algorithm to improve its performance towards a goal. Unlike previous work, Reinforcement Teaching can be applied across different problem settings.

2. Rather than having the teacher learn directly from the student's parameters, a *parametric-behavior embedder* learns a state representation from the student's inputs and outputs. This provides a problem-agnostic state representation that improves the teacher's learning, and allows Reinforcement Teaching with deep non-linear students.

3. We define a *learning progress* reward function that further accelerates learning by improving the teacher's credit assignment.

To demonstrate the generality and effectiveness of Reinforcement Teaching, we apply this framework, with the parametric-behavior embedded state and learning progress reward, in two domains (1) curriculum learning and (2) step-size adaptation. Results in discrete and continuous control environments show examples of

Reinforcement Teaching, in which the teacher learns a policy that selects sub-tasks for an RL student. In step-size adaptation for supervised learning students, we show that a reinforcement teacher can learn a policy that adapts the step-size of Adam (Kingma & Ba, 2015), improving upon the best constant step-size. Moreover, in both settings our Reinforcement Teaching method learns a superior teaching policy, compared to several other baselines, that results in improved student learning. The primary goal of this paper is to spur novel developments in meta-learning using the tools of RL, and to unify different RL-based approaches under the single framework of Reinforcement Teaching.

## 2 Sequential Decision Making for Meta-learning

Before introducing Reinforcement Teaching, we argue for the importance of a sequential decision making perspective for meta-learning. Reinforcement Teaching, presented in Section 4, develops a framework that allows reinforcement learning algorithms to be applied in sequential meta-learning settings.

Many meta-learning settings are sequential, such as hyper-parameter adaptation, curriculum learning, and learned optimization. While there are many meta-learning settings of interest, we use step-size adaptation as an illustrative example because of its history and ubiquity in the meta-learning literature (Schraudolph, 1999; Maclaurin et al., 2015; Sutton, 1992; 2022; Kearney et al., 2018). Step-size adaptation is the problem of selecting a learning rate at each step of an optimization algorithm. This is different from hyperparameter optimization, or tuning, where a grid-search is used to select the best constant learning rate for all steps of the optimization algorithm. In particular, we will use the noisy-quadratic problem studied in Schaul et al. (2013) and Wu et al. (2018b), in which a learner with parameters $\theta$ attempts to minimize an objective with a stochastic minimum. This problem, while simple, is an illustrative example of the importance of step-size adaptation in stochastic settings. The objective function, defined for a d-dimensional parameter vector, $\theta = (\theta_1, \ldots, \theta_d)$, depends on a stochastic variable, $c = (c_1, \ldots, c_d)$, determining the minimum and a fixed diagonal hessian with entries, $h = (h_1, \ldots, h_d)$. If we assume that the stochastic minimum follows an independent Gaussian distribution, $c_i = \mathcal{N}(0, \sigma_i)$, then we can write the expected value of the objective as, $\mathcal{L}(\theta) = \mathbb{E}[\hat{\mathcal{L}}(\theta)] = \mathbb{E}\left[\frac{1}{2}\sum_{i=1}^{d} h_i(\theta_i - c_i)^2\right] = \frac{1}{2}\sum_{i=1}^{d} h_i\left(\mathbb{E}[\theta_i]^2 + \mathbb{V}[\theta_i] + \sigma_i^2\right)$. The learner uses gradient descent (with or without momentum) to update its parameters from $\theta^{(t-1)}$ to $\theta^{(t)}$ with a step-size of $\alpha^{(t)}$. In this setting, the stochastic gradient is $\frac{\partial \hat{\mathcal{L}}}{\partial \theta_i} = h_i(\theta_i - c_i)$ and the deterministic gradient is $\frac{\partial \mathcal{L}}{\partial \theta_i} = h_i\theta_i$. Meta-learning in the noisy quadratic problem amounts to selecting each $\alpha^{(t)}$ such that the learnable parameters after $T$ steps of gradient descent best minimizes the objective, given by $\mathcal{L}(\theta^{(T)})$.

There are two strategies for meta-learning $\alpha^{(t)}$: fully-optimized or one-step greedy-optimal. The fully-optimized sequence of step-sizes $\{\alpha^{(t)}\}_{t=1}^{T}$ is jointly chosen to minimize the loss at step $T$, where the loss is given by $\mathcal{L}(\theta^{(T)})$. Even in the simple noisy quadratic problem, fully-optimizing the step-size can be computationally costly. An alternative is the one-step greedy-optimal schedule which selects $\alpha^{(t)}$ so as to minimize the loss at the next iteration, $\mathcal{L}(\theta^{(t)})$. In either the deterministic-gradient or spherical-gradient setting (where all entries of $h$ are the same, see Wu et al. (2018b), Theorem 3), the one-step greedy-optimal step-size coincides with the fully-optimized schedule. In general, however, the fully-optimized step-size schedule can result in a much lower final loss compared to one-step greedy-optimal schedule.

If we consider the step-size as the meta-learner's action, $a_t = \alpha^{(t)}$, the one-step greedy-optimal strategy treats noisy quadratic optimization as a contextual bandit problem where the state is the current parameter, $s_t = \theta^{(t)}$. At each time-step, the action is selected so as to minimize the next immediate loss. In the context of rewards, we may define the reward as $r_t = -\mathcal{L}(\theta^{(t)})$. The fully optimized sequence of $\{\alpha^{(t)}\}_{t=1}^{T}$ can also be thought of as a bandit problem where the action is the joint selection of $\{\alpha^{(t)}\}_{t=1}^{T}$. This is costly, requiring re-computation of every parameter iterate, $\{\theta^{(t)}\}_{t=1}^{T}$, for each candidate step-size sequence. A more natural formulation to learning the fully-optimized schedule can use reinforcement learning, where at each time-step a policy observes the learnable parameters, $\theta^{(t-1)}$, and selects a step-size, $a_t = \alpha^{(t)}$, so as to minimize the long-term loss. There are many reward functions that incentive the policy to minimize the long-term loss, such as a finite-horizon terminal reward ($r_{t<T} = 0, r_T = -\mathcal{L}(\theta^{(T)})$), or having a reward of $-1$ until a loss threshold, $\mathcal{L}^*$, is reached which then terminates the episode ($r_t = -\mathbb{I}(\mathcal{L}(\theta^{(t)}) > \mathcal{L}^*)$, $\mathbb{I}(x > y) = 1$ if $x > y$). This reinforcement perspective is not specific to step-size adaptation; we develop Reinforcement Teaching in Section 4 for any sequential meta-learning problem.

## 3 Related Work

**Learning to Teach Using Reinforcement Learning**  Using an RL teacher to control particular aspects of another student's learning process has been previously explored (Almeida et al., 2021; Garcia & Thomas, 2019; Huang et al., 2019; Cubuk et al., 2019; Ruiz et al., 2019; Wu et al., 2018a; Dennis et al., 2020; Campero et al., 2020; Florensa et al., 2018; Fan et al., 2018; Narvekar et al., 2017; Narvekar & Stone, 2019; Zhu et al., 2019; Zoph & Le, 2017; Jomaa et al., 2019; Biedenkapp et al., 2020; Sabbioni et al., 2020).

However, by the design of these solution methods, they are only suitable for solving specific meta-learning problems and lack applicability across different learning problems. More specifically, these works use problem-specific heuristics to construct the teacher's state representation that results in non-Markov state representations (Wu et al., 2018a; Dennis et al., 2020; Campero et al., 2020; Florensa et al., 2018; Fan et al., 2018; Huang et al., 2019; Jomaa et al., 2019; Zhu et al., 2019). This is commonly done because the Markov state representation, the student's parameters, is a large and unstructured state representation that makes it difficult to learn an effective policy. As a representative of the heuristic approach, the L2T framework (Fan et al., 2018) successfully learned to sample minibatches for a supervised learner. In this approach, the teacher's state representation includes several heuristics about the data and student model and is heavily designed for the task of minibatch sampling (Fan et al., 2018). These works are tailored to the base problems they solve and are unable to generalize to new problems with their state and reward design. Some works have identified that learning from parameters is theoretically ideal for curriculum learning (Narvekar et al., 2017). However, the success of parameter state representation has been limited to either toy problems (e.g., 1D regression) or tabular/linear RL students (Biedenkapp et al., 2020; Zhang et al., 2020; Narvekar et al., 2017; Sabbioni et al., 2020). Until now, no work has attempted to use the behavioral approach proposed in this paper to enable tractable, generalizable, and transferable learning algorithms.

These approaches can be contrasted bandit formulations of the student-teacher setting (Portelas et al., 2019; Graves et al., 2017; Jiang et al., 2021a; Parker-Holder et al., 2022; Jiang et al., 2021b). Although the bandit formulation has demonstrated promising results in the automatic curriculum learning domain, it can be limiting for other meta-learning problems such as step-size adaptation (See Section 2).

**Learning Progress**  Connected to the idea of teaching is a rich literature on learning progress. Learning progress prescribes that a learning agent should focus on tasks for which it can improve on. This mechanism drives the agent to learn easier tasks first, before incrementally learning tasks of increasing complexity (Oudeyer et al., 2007). Learning progress has been represented in several ways such as the change in model loss, model complexity, and prediction accuracy. In addition, learning progress has been successfully applied in a variety of contexts, including curriculum learning (Portelas et al., 2019; Oudeyer et al., 2007; Matiisen et al., 2020; Graves et al., 2017), developmental robotics (Blank et al., 2003; Moulin-Frier Clément, 2014; Oudeyer et al., 2007), and intelligent tutoring systems (Clement et al., 2015).

**Learned Parameter Representations**  Previous work in RL has argued that policies can be represented by a concatenated set of outputs (Harb et al., 2020; Parker-Holder et al., 2020). Policy eValuation Networks (PVN) in RL show that representations of a neural policy can be learned through the concatenated outputs of a set of learned inputs. PVN is similar to the parametric-behavior embedder that we propose because it characterizes a neural network by its output behavior. Learning a PVN representation, however, requires a fixed set of inputs, referred to as probing inputs. While the probing inputs can be learned, they are still fixed after learning and cannot adapt to different policies. In our setting, the student's neural network is frequently changing due to parameter updates and it is unlikely that the outputs of a fixed set of inputs can represent the changing parameters during learning. Furthermore, Faccio et al. (2021) showed that learning to evaluate policies directly from parameters is more performant than PVNs for policy improvement, suggesting that fixed probing inputs are insufficient for representing many neural networks.

**Reward Design**  In standard reinforcement learning, an agent's goal is to maximize the expected sum of a discounted scalar reward signal. However, the source of this reward is unspecified and it is typically left to a designer to craft the agent's reward function. As reward design is a non-trivial task, past work has cast the problem of designing the reward function as an optimization problem – the optimal reward problem (Singh

et al., 2009; Sorg et al., 2010; Jain et al., 2021). Another related sub-area is adaptive reward shaping, in which an RL teacher agent learns to adaptively shape the student's reward function (Mguni et al., 2023). Reward design can be seen as a special case of Reinforcement Teaching. From this perspective, the RL teacher would take actions that adjust the student's reward function during the student's learning process to improve their overall learning.

**Machine Teaching** Machine teaching is a general paradigm in which a teacher is used to guide a student. A widely studied application of machine teaching is the supervised learning setting in which a teacher is tasked with choosing the best training set such that a machine learning student can learn a target model (Zhu et al., 2018). Recent work has applied machine teaching to sequential decision-making tasks. In this setting, machine teaching has been used to study a wide range of problems, from finding the best set of demonstrations to finding the best reward shaping strategy (Brown & Niekum, 2019; Zhang et al., 2020).

Under the Reinforcement Teaching perspective, machine teaching can be viewed as an RL teacher whose action determines the data that the student uses for learning. The primary issue with traditional machine teaching approaches is that they assume the teacher has access to an optimal student model, learning algorithm, and objective function (Zhu et al., 2018). These assumptions are unrealistic in practice. We show that our parametric-behavior embedded state and learning progress reward allows the teacher to learn a policy while only having access to the student's inputs/outputs and performance.

**Meta-Learning** While Reinforcement Teaching does not explicitly build on previous meta-learning work, we point out common meta-learning methods and how they relate to Reinforcement Teaching. Early work in meta-learning with neural networks (Younger et al., 2001; Hochreiter et al., 2001; Schmidhuber, 1987; Sutton, 1992) inspired follow-up work on learned optimizers (Ravi & Larochelle, 2017; Andrychowicz et al., 2016). Learned optimizers replace the fixed learning algorithm with a memory-based parameterization, usually an LSTM (Hochreiter & Schmidhuber, 1997). Learning the optimizer through reinforcement learning has also been explored (Li & Malik, 2017a;b). This work, like the approach by Fan et al. (2018), employs an ad-hoc state representation and reward function. Optimization-based meta-learning has other applications, such as in few-shot learning (Ravi & Larochelle, 2017) and meta-RL (Duan et al., 2016; Wang et al., 2016). Another approach to meta-learning is gradient-based meta-learning, such as Model Agnostic Meta Learning (MAML) (Finn et al., 2017) and other work in meta-RL (Xu et al., 2018). These methods are distinguished from optimization-based meta-learning for the lack of a separately parameterized meta-learner. Instead, meta-information is encoded in $\theta$ by differentiating through gradient descent.

## 4 Reinforcement Teaching

Before introducing Reinforcement Teaching, we first describe the MDP formalism that underpins reinforcement learning (Lattimore & Szepesvári, 2020; Sutton & Barto, 2018; Puterman, 2014). An MDP $\mathcal{M}$ is defined by the tuple $(\mathcal{S}, \mathcal{A}, r, p, \mu, \gamma)$, where $\mathcal{S}$ is the state space, $\mathcal{A}$ denotes the action space, $\mathcal{S}$ is the state space, $r : \mathcal{A} \times \mathcal{S} \rightarrow \mathbb{R}$ is the reward function that maps a state and an action to a scalar reward, $p : \mathcal{S} \times \mathcal{A} \times \mathcal{S} \rightarrow [0, 1]$ is the state transition function, $\mu$ is the initial state distribution, and $\gamma$ is the discount factor. Lastly, a Markov reward process (MRP) is an MDP without actions (Sutton & Barto, 2018). For an MRP, both the reward function $r : \mathcal{S} \rightarrow \mathbb{R}$ and state transition $p : \mathcal{S} \times \mathcal{S} \rightarrow [0, 1]$ are no longer explicitly a function of an action. Instead, actions are unobserved and selected by some unknown behavior policy.

In Reinforcement Teaching, *student* refers to any learning agent or machine learning model, and *teacher* refers to an RL agent whose role is to adapt to and improve the student's learning process. We start by defining the components of the student's learning process. We then identify states and rewards, thereby formulating the student's learning process as an MRP. This MRP perspective on learning processes allows the Reinforcement Teaching framework to be applied to different types of students with varying data domains, learning algorithms, and goals. Lastly, we introduce an action set for the teacher which allows the teacher to alter the student's learning process. This induces an MDP, in which the teacher learns a policy that interacts with a student's learning process to achieve a goal (see Figure 1).

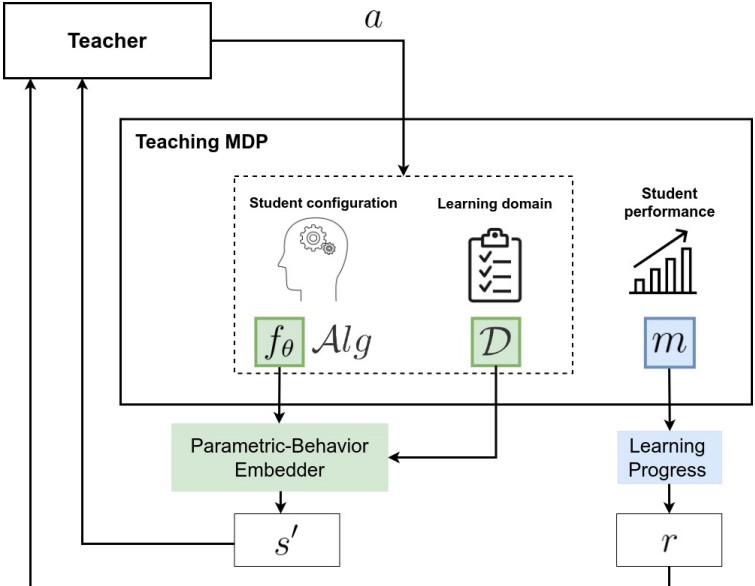

Figure 1: The teacher takes actions $a \in \mathcal{A}$, which will influence an aspect of the teaching MDP, such as the student, $f_\theta$, learning algorithm, $\mathcal{A}lg$, or learning domain $\mathcal{D}$. The student will then update its parameters, $\theta$, and the teaching MDP will then output $r, s'$ based on the student's new parameters.

### 4.1 Components of the Learning Process

To start, we define the student learning process and its components. Consider a student, $f_\theta$, with learnable parameters $\theta \in \Theta$. The student receives experience from a learning domain $\mathcal{D}$, which can be labeled data (supervised learning), unlabelled data (unsupervised learning), or an MDP (reinforcement learning). How the student interacts with, and learns, in a domain is specified by the student's learning algorithm $\mathcal{A}lg$. The student's learning algorithm updates the student's parameters, $\theta_{t+1} \sim \mathcal{A}lg(f_{\theta_t}, \mathcal{D})$, in order to maximize a performance measure that evaluates the student's current ability, $m(f_\theta, \mathcal{D})$. Written this way, $m$ can be seen as the objective function directly optimized by $\mathcal{A}lg$, but $m$ can also be a non-differentiable metric such as accuracy in classification, or the Monte-Carlo return in RL.

The combination of the student, learning domain, learning algorithm, and performance measure is hereafter referred to as the student's learning process, $\mathcal{E} = (f_\theta, \mathcal{D}, \mathcal{A}lg, m)$. In the remainder of Section 4, we will outline how the components of the learning process interact as the student learns the optimal parameters that maximize its performance measure, $\theta^* = \arg\max_\theta m(f_\theta, \mathcal{D})$.

### 4.2 States of Reinforcement Teaching

We define the state of the learning process as the student's current learnable parameters, $s_t = \theta_t$. Therefore, the state space is the set of possible parameters, $\mathcal{S} = \Theta$. The initial state distribution, $\mu$, is determined by the initialization method of the parameters, such as Glorot initialization for neural networks (Glorot & Bengio, 2010). Lastly, the state transitions, $p$, are defined through the learning algorithm, $\theta_{t+1} = \mathcal{A}lg(f_{\theta_t}, \mathcal{D})$, which can be stochastic in general.

The sequence of learnable parameters, $\{\theta_t\}_{t \geq 0}$, form a Markov chain as long as $\mathcal{D}$ and $\mathcal{A}lg$ do not maintain their own state that depends on the parameter history. This is the case, for example, when the learning domain is a dataset[1], $\mathcal{D} = \{x_i, y_i\}_{i=1}^N$, and the learning algorithm is gradient descent on an objective function, $\theta' := \mathcal{A}lg(f_\theta, \mathcal{D}) = \theta - \alpha \nabla_\theta \frac{1}{N} \sum_{i=1}^N J(f_\theta(x_i), y_i)$ (Mandt et al., 2017; Dieuleveut et al., 2020). While adaptive optimizers violate the Markov property of $\mathcal{A}lg$, we discuss ways to remedy this issue in Appendix D and demonstrate that it is possible to learn a policy that controls Adam (Kingma & Ba, 2015) in Section 5.2.

---

[1]RL environments are also Markovian learning domains if the environment itself is Markovian.

### 4.2.1 Parametric-behavior Embedder

Although $\theta$ is a Markov state representation, it is not ideal for learning a policy. To start, the parameter space is large and mostly unstructured, especially for nonlinear function approximators. While there is some structure and symmetry to the weight matrices of neural networks (Brea et al., 2019; Fort & Jastrzebski, 2019), this information cannot be readily encoded as an inductive bias of a meta-learning architecture. Often, the parameter set is de-structured through flattening and concatenation, further obfuscating any potential regularities in the parameter space. Ideally, the teacher's state representation should be much smaller than the parameters. As smaller state spaces simplify the learning problem on behalf of the teacher. In addition, the teacher's state representation should allow for generalization to new student models with different architectures or activations, which is not feasible with the parameter state representation. With this property, the teacher does not have to learn a separate teaching policy for each type of student model. See Section 5.2 for empirical evidence of the difficulty of learning from parameters.

To avoid learning from the parameters directly, we propose the *parametric-behavior embedder* (PE), a novel method that learns a representation of the student's parameters from the student's behavior. To capture the student's behavior, we use the inputs and outputs of $f_\theta$, as well as the targets for the student. For example, if the student is a classifier, the inputs to $f_\theta$ would be the features $x_i$, the targets would be the label $y_i$, and the outputs would be the classifier's predictions, $f_\theta(x_i)$. To learn the PE state representation, we first assume that we have a dataset or replay buffer to obtain the student inputs and targets. Then we can randomly sample a minibatch of $M$ inputs, $\{x_i, y_i\}_{i=1}^M$, and retrieve the student's corresponding outputs, $f_\theta(x_i)$. The set of inputs, targets and student outputs $\hat{s} = \{x_i, y_i, f_\theta(x_i)\}_{i=1}^M$, or mini-state, provides local information about the true underlying state $s = \theta$. To learn a vectorized representation from the mini-state, we recognize that $\hat{s}$ is a set and use a permutation invariant function $h$ to provide the PE state representation $h(\hat{s})$ (Zaheer et al., 2017). The input-output pair is jointly encoded before pooling, $h(\hat{s}) = h_{pool}\left(\{h_{joint}(x_i, y_i, f_\theta(x_i))\}_{i=1}^M\right)$, where $h_{pool}$ is a pooling operation over the minibatch dimension (see Figure 2).

We argue that the parametric-behavior embedder approximates the Markov state $\theta$. This state representation uses local information provided by the student's behavior. With a large enough minibatch of inputs and outputs, it can summarize pertinent information about the current $\theta$ and how it will change, thereby approximating the Markov state (See Appendix F for more details). Methods that attempt to learn directly from the parameters must learn to ignore aspects of the parameters that have no bearing on the student's progress. This is inefficient for even modest neural networks. As we demonstrate in Section 5, the PE state representation allows the teacher to learn an effective teaching policy compared to several other baselines.

## 4.3 Rewards of Reinforcement Teaching

Given a reward function, $r$, we further formalize the learning process as an MRP, $\mathcal{E} = (\mathcal{S}, r, p, \mu)$, where the state-space $(\mathcal{S})$, initial distribution $(\mu)$, and state-transition dynamics $(p)$ are defined in Section 4.2. The learning process is formalized as an MRP for two reasons: (1) learning processes are inherently sequential, and therefore an MRP is a natural way to depict the evolution of the student's parameters and performance, and (2) MRPs provide a unifying framework for different students' algorithms and learning domains.

To specify the reward function, we first identify that reaching a high-level of performance is a common criterion for training and measuring a learner's performance.[2] For ease of reference, let $m(\theta) := m(f_\theta, \mathcal{D})$. A simple approach is the time-to-threshold reward in which a learner is trained until a performance condition is reached, such as a sufficiently high performance measure (i.e., $m(\theta) \geq m^*$ for some threshold $m^*$) (Narvekar et al., 2017). In this case, the reward is constant $r(\theta) = -\mathbb{I}(m(\theta) < m^*)$ until the condition, $m(\theta) \geq m^*$, is reached, which then terminates the episode.

Similar to the argument in Section 4.2, the reward function $r(\theta) = -\mathbb{I}(m(f_\theta, \mathcal{D}) < m^*)$ is also Markov as long as the learning domain is Markov. The performance measure itself is always Markov because, by definition, it evaluates the student's current ability.

---

[2] Appendix C outlines alternative reward criteria and reward shaping in the Teaching MRP.

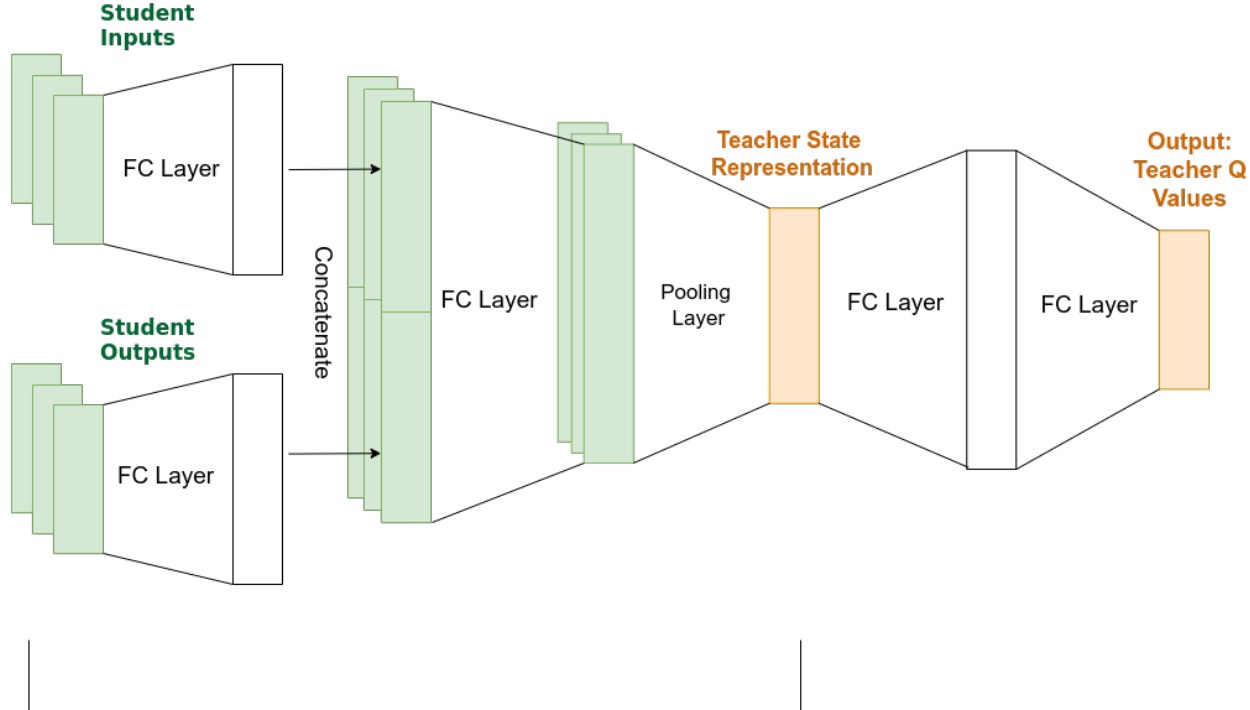

Figure 2: The neural network architecture used for Reinforcement Teaching with the parametric-behavior embedding state representation. For a given student, $f_\theta$, the parametric-behavior embedder independently projects a mini-batch of student inputs, $\{x_i\}_{i=1}^M$, and student outputs, $\{f_\theta(x_i)\}_{i=1}^M$, into a latent space before concatenation and pooling, providing a state representation of $\theta$.

### 4.3.1 Reward Shaping with Learning Progress

Under the time-to-threshold reward (Narvekar et al., 2017), the teacher is rewarded for taking actions such that the student reaches a performance threshold $m^*$ as quickly as possible. We argue, however, that this binary reward formulation lacks integral information about the student's learning process.

To address this shortcoming, we define a new reward function based on the student's *learning progress*. The learning progress signal provides feedback about the student's relative improvement and better informs the teacher about how its policy influences the student.

We define Learning Progress (LP) as the change in the student's performance measure, $LP(\theta', \theta) = m(\theta') - m(\theta)$ at subsequent states $\theta$ and $\theta'$ of the student's learning process. To shape the time-to-threshold reward, we add the learning progress term $LP(\theta', \theta)$ to the existing reward $r(\theta')$ previously described. Therefore, our resulting LP reward function is $r(\theta', \theta) = -\mathbb{I}(m(\theta) < m^*) + LP(\theta', \theta)$ until $m(\theta) \geq m^*$, terminating the episode. It follows that learning progress is a potential-based reward shaping, given by $r' = r + \Phi(\theta') - \Phi(\theta)$, where the potential is the performance measure $\Phi(\theta) = m(\theta)$. This means that combining learning progress with the time-to-threshold reward does not change the optimal policy (Ng et al., 1999).

Unlike the time-to-threshold reward function, the LP reward provides critical information to the teacher regarding how its actions affected the student's performance. The LP term indicates the extent to which the teacher's adjustment (i.e., action) improved or worsened the student's performance. For example, if the teacher's action results in a negative LP term, this informs the teacher that with the student's current skill level (as defined by the student's parameters), this specific action worsened the student's performance, thereby deterring the teacher from selecting such an action. We show empirically that compared to the

time-to-threshold reward and other reward functions found in the literature, the LP reward function enables the teacher to learn a more effective teaching policy (See Section 5.1, Figure 4 and Table 3).

### 4.4   Actions of Reinforcement Teaching

The MRP model demonstrates how the student's learning process can be viewed as a sequence of parameters, $\{\theta_t\}_{t \geq 0}$, with rewards describing the student's performance at particular points in time, $\{m(\theta_t)\}_{t > 0}$. However, the goal of meta-learning is to improve this learning process. The teacher now oversees the student's learning process and takes actions that intervene on this process, thus transforming the MRP into the Teaching MDP, $\mathcal{M} = (\mathcal{S}, \mathcal{A}, p, r, \mu)$. Aside from the action space, $\mathcal{A}$, the remaining elements of the Teaching MDP tuple have been defined in the previous subsections.

We now introduce the action set, $\mathcal{A}$, that enables the teacher to control some component of the student learning process. An action can change the student configuration or learning domain of the student, as shown in Figure 1. Similar to RL environments, we take the action set as part of the meta-learning task description and do not make further assumptions about the role of the action. The choice of action space induces different meta-learning problem instances (see Appendix B), such as learning to sample, learning to explore, curriculum learning (learning a policy for sequencing tasks), and adaptive optimization (learning to adapt the step-size).

Lastly, the action set determines the time-step of the teaching MDP. The base time-step is each application of $\mathcal{A}lg$, which updates the student's parameters. The teacher can operate at this frequency in settings where it controls an aspect of the learning algorithm, such as the step-size. In this setting, the teacher would take an action (e.g., select a step-size) after every parameter update for the student. Acting at a slower rate induces a semi-MDP (Sutton et al., 1999). If the teacher controls the learning domain, such as setting an episodic goal for an RL agent, then the teacher could operate at a slower rate than the base time-step. This would result in the teacher taking an action (e.g., selecting a goal) after a complete student episode(s) which comprises several student parameter updates. With the full Reinforcement Teaching framework outlined, see Algorithm 1 for the corresponding pseudocode of the teacher-student interaction.

---

**Algorithm 1** Reinforcement Teaching Framework

---

**Input**: teacher RL algorithm $\psi^T$, student ML algorithm $\mathcal{A}lg$, replay buffer D for student inputs/outputs, teacher action set $\mathcal{A}$, initial teacher parameters $\theta_T$, learning domain $\mathcal{D}$, and minibatch size $M$ and student performance threshold $m^* \in [0, 1]$

**Loop** for each teacher episode:

    Reset student parameters $\theta_s$ and $m(\theta_s) = 0$

    Set initial teacher state $S$

    **While** $m(\theta^s) < m^*$ **do**:

        Choose teacher action $A \in \mathcal{A}$ and update the student's learning process $\mathcal{E}$

        Train student via $\mathcal{A}lg$. During this training store student inputs $x$ in D

        Randomly sample a minibatch of $M$ inputs from D, $\{x_i\}_{i=1}^M$

        Retrieve the student's corresponding outputs to obtain $\{x_i, f_{\theta_s}(x_i)\}_{i=1}^M$

        Calculate $S' = h_{pool}\left(\{h_{joint}(x_i, f_\theta(x_i))\}_{i=1}^M\right)$

        Evaluate student on learning domain $\mathcal{D}$ to obtain $m(\theta_s')$

        Calculate $LP = m(\theta_s') - m(\theta_s)$

        Calculate $R' = -\mathbb{I}\left(m(\theta_s') < m^*\right) + LP$

        Update $\theta^T$ according to $\psi^T$

---

## 5   Experiments

To demonstrate the generality and effectiveness of Reinforcement Teaching, we conduct experiments in both curriculum learning (Section 5.1) and step-size adaptation (Section 5.2).

|  |  | Teacher Action | # of Teacher Actions | Frequency of Teacher Action | Teacher State | Teacher Reward |
|---|---|---|---|---|---|---|
| Curriculum Learning | Maze | Start state | 11 | After complete student episode(s) | PE variants | LP |
|  | Four Rooms | Start state | 10 | After complete student episode(s) | PE variants | LP |
|  | Fetch Reach | Goal distribution | 9 | After complete student episode(s) | PE variants | LP |
| Step-size Adaption | Synthetic Classification with SGD | Relative change in step-size | 3 | After each student gradient step | PE variants | LP |
|  | Synthetic Classification with Adam | Relative change in step-size | 3 | After each student gradient step | PE variants | LP |

Table 1: Initialization of teaching MDP for the Curriculum Learning and Step-size adaption problem settings.

In the curriculum learning setting, we show that the teacher using the PE state representation and LP reward function significantly outperforms other RL teaching baselines in both discrete and continuous environments. For the step-size adaptation setting, we show that only the PE state representation can learn a step-size adaptation policy that improves over Adam with the best constant step-size. We further show that this step-size adapting teacher learns a policy that generalizes to new architectures and datasets. Our results confirm that both PE state and LP reward are critical for Reinforcement Teaching, and significantly improves over baselines that use heuristic state representations and other parameter representations.

## 5.1   Curriculum Learning For Reinforcement Learning Students

In this section, we apply our Reinforcement Teaching framework to the curriculum learning problem. Our goal is for the teacher to learn a policy for sequencing sub-tasks such that the student can solve a target task quickly. In our experiments, we consider both discrete and continuous environments: an 11 by 16 tabular maze, Four Rooms adapted from the MiniGrid suite (Chevalier-Boisvert et al., 2018), and Fetch Reach (Plappert et al., 2018). For the maze and Four Rooms environment, the student's objective is to learn the most efficient path from a target start state to a target terminal state. For the Fetch Reach environment, the student's goal is to learn how to move the end-effector to random locations in 3D space, given a fixed start state. See Appendix I.1 for more details on the student environments.

**Teaching MDP for Curriculum Learning**   To formalize curriculum learning through Reinforcement Teaching, we establish the teaching MDP (see Table 1). We begin by discussing the teacher's action space. The teacher's actions will control an aspect of the student's environment. For the maze and Four Rooms environment, the teacher's action will change the student's start state. The teacher can select the student's initial position from a pre-determined set of states, which can include states that are completely blocked off.

For Fetch Reach, the teacher's actions determine the goal distribution. The goal distribution determines the location the goal is randomly sampled from. Each action gradually increases the maximum distance between the goal distribution and the starting configuration of the end-effector. Therefore, "easier" student sub-tasks are ones in which the set of goals are very close to the starting configuration. Conversely, "harder" tasks are ones in which the set of goals are far from the starting configuration of the end-effector.

For the teacher's state representation, we consider two variants of PE that use different student outputs $f_\theta$. In both cases, the inputs are the states that the student encounters during its learning process (i.e., student training episodes). For PE-Values, the embedded outputs are the state/state-action values, whereas for PE-Action, the embedded outputs are the student's actions. Specifically, during each student episode, the student encounters (state, action) pairs. These pairs are then stored in a buffer. If the student state is already in the buffer, we keep the latest action that was taken. When it's time to retrieve the teacher's state representation, we randomly sample a minibatch of M (state, action) pairs from this buffer. For the PE-Values representation, we query the state/action value corresponding to each state in the minibatch using the most up-to-date value network. Finally, for all reward functions, the performance measure is the student's return on the target task.

For the student's learning algorithm, we used Q learning (Sutton & Barto, 2018), PPO (Schulman et al., 2017), and DDPG (Lillicrap et al., 2016) for the maze, Four Rooms, and Fetch Reach environments, respectively. This highlights that Reinforcement Teaching can be useful for a variety of students. See Table 10 for full specification of student hyperparameters.

**Teacher Training**   Now, to train the teacher, we use DQN (Mnih et al., 2015). We use DQN for two reasons. First, we wanted our Reinforcement Teaching framework to have low sample complexity. As

a single teacher episode corresponds to an entire training trajectory for the student, generating numerous teacher episodes involves training numerous students. The teacher agent cannot afford an inordinate amount of interaction with the student. One way to meet the sample complexity needs of the teacher is to use off-policy learning, such as Q-learning. Off-policy learning is generally more sample efficient than on-policy methods because of its ability to reuse past experiences that are stored in the replay buffer. Therefore, the family of DQN algorithms is one natural choice. Second, our goal is to evaluate the efficacy of our Reinforcement Teaching framework in solving multiple meta-learning problems. Although there have been advancements in off-policy learning algorithms (Lillicrap et al., 2016; Haarnoja et al., 2018; Fujimoto et al., 2018) and these improvements are likely to improve the performance of our framework, we wanted to study Reinforcement Teaching in the simplest deep RL setting possible.

We follow the pseudocode in Algorithm 1 to train the teacher. See Appendix J for full details on teacher hyperparameters.

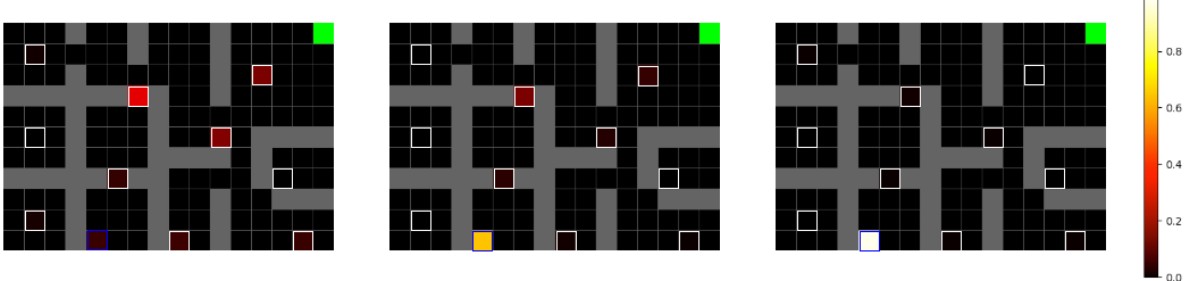

Figure 3: The beginning (left), middle (center), and ending (right) stages of the curriculum generated by the PE-Actions + LP method for the Maze environment. States outlined in white indicate possible teacher actions. The state outlined in blue indicates the target start state and the green state indicates the target goal state. Brighter colors (more yellow/white) indicate the start state was chosen more frequently by the teacher. Darker red/black indicates the start state was chosen less frequently by the teacher.

**Teacher Evaluation** To evaluate the teacher's policy, we follow a similar protocol as done in training. The teacher's policy is first frozen, and the teacher is assigned a single newly initialized student. The teacher then interacts with this student by taking actions (e.g., providing sub-tasks) that are provided to the student. During the teacher evaluation, the goal is to determine whether the teacher provides a curriculum of sub-tasks to the student such that the student can learn its target task efficiently. To show this, we report the student's learning curves (while using the teacher's curriculum) in Figure 4. To analyze the effectiveness of the PE state and the LP reward function on the teacher's policy, we compare against the following RL teaching baselines: L2T (Fan et al., 2018) and Narvekar et al. (2017). Narvekar et al. (2017) uses the parameter state representation with the time-to-threshold reward. Fan et al. (2018) uses a heuristic-based state representation and a variant of the time-to-threshold reward. We also compare against TSCL Online (Matiisen et al., 2020), a representative of the multi-armed bandit approaches from the curriculum learning literature, a random teacher policy, and a student learning the target task from scratch (no teacher). Moreover, as a typical consequence of using RL to train a teacher is the additional training computation, we also compare the teacher's own learning efficiency across the RL-teaching methods (see Figure 4-left). These results indicate that with our method, the teacher can learn effective curricula more quickly than existing RL teaching baselines. All results are averaged over 30 seeds with shaded regions indicating 95 % confidence intervals (CI).

**Experimental Results** Across all environments, we found that by using either of our PE state representations along with our LP reward signal, the teacher is able to learn a comparable or superior curriculum policy compared to the baselines. These teacher policies generated a curriculum of start/goal states for the student that improved the student's learning efficiency and/or final performance, as shown in Figure 4-right. For example, we found that in the Maze domain, the PE-Actions + LP teacher policy initially selected

starting states close to the target goal state. However, as the student's skill set improved over time, the teacher adapted its policy and selected starting states farther away from the goal state (see Figure 3).

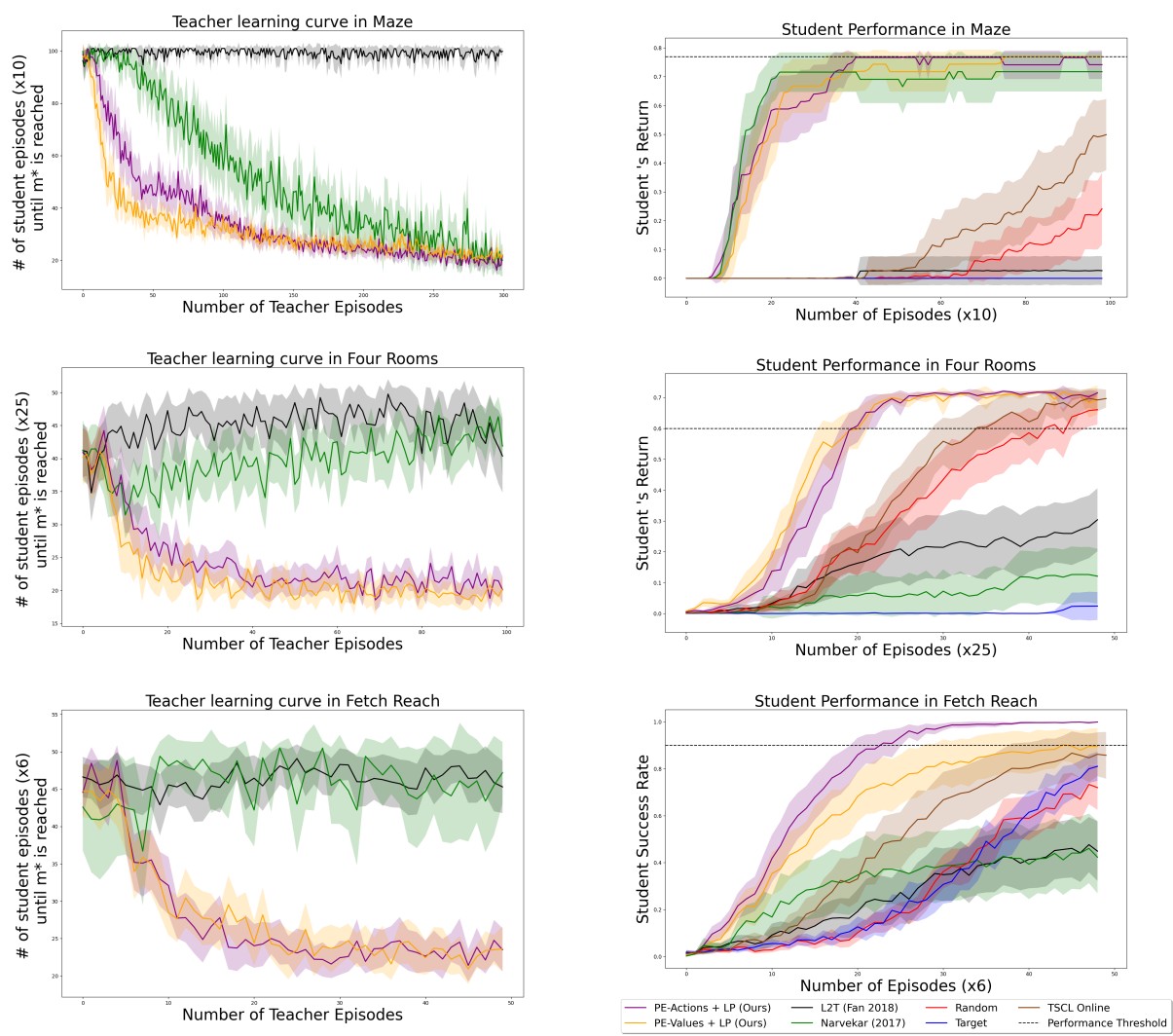

Figure 4: The left plots are learning curves for the teacher. The y-axis is the number of episodes needed for the student to reach the performance threshold, $m^*$, with the teacher's current policy, as the teacher learns over episodes on the x-axis (lower is better). The right plots are the student's training curves while using the trained teacher's curriculum policy (higher is better).

Moreover, in the Maze domain, we found that the teacher was able to learn a comparable policy using the Narvekar et al. (2017) baseline. This is not surprising because in this domain the student's parameters are represented by the tabular action-value table. This parameter set is small and does not come with the same issues as the parameters of a function approximator as described in Section 4.2.

However, only our method is able to maintain significant improvements in student learning even as the student environment becomes more complex as demonstrated by Four Rooms and Fetch Reach results. Lastly, we found that with our approach, the teacher is able to learn these curriculum policies efficiently compared to the other RL-teaching baselines (See Figure 4-left). This is important because RL-teaching approaches, like Reinforcement Teaching, require computation on behalf of both the teacher and student algorithm. Therefore, it is crucial that our framework learns effective policies as quickly as possible.

**Ablating State and Reward Functions**  To highlight the importance of our state representation and reward function on the teacher's learned policy, we ablate over various state representations and reward functions used in the literature. We report the area under the student's learning curve (AUC) when trained using the teacher's learned curriculum (See Tables 2 and 3). We use a one-tailed independent-samples Welch t-test (*i.e.,* equal variances are not assumed) to determine if there is a difference in the average AUC between methods with a p-value of 0.05.[3]

### State ablation

|  | PE-Value (Ours) | PE-Action (Ours) | L2T (Fan 2018) | Parameters (Narvekar 2017) |
|---|---|---|---|---|
| Maze | $62.12 \pm 1.73$ | $61.62 \pm 1.90$ | $61.44 \pm 2.51$ | $\mathbf{66.62 \pm 0.96}$ |
| Four Rooms | $\mathbf{25.33 \pm 0.56}$ | $22.98 \pm 0.76$ | $25.18 \pm 1.10$ | $6.0 \pm 2.94^{**}$ |
| Fetch Reach | $29.72 \pm 2.95$ | $\mathbf{34.76 \pm 1.94}$ | $29.75 \pm 1.56^{*}$ | $16.13 \pm 4.54^{**}$ |

Table 2: Ablation of teacher state representation functions with fixed LP reward function. Reporting mean area under the student's learning curve plus/minus standard error. The results are over 10 runs. * Indicates a significant difference (p<.05) between our PE state representation and the baseline representations.  ** Indicates a significant difference between baseline and both of our state representations (PE Values/Actions). Bold indicates the highest mean area under the curve.

### Reward ablation

|  |  | LP (Ours) | Time-to-threshold | L2T reward | Ruiz (2019) reward | Matiisen (2020) reward |
|---|---|---|---|---|---|---|
| Maze | PE-Value (Ours) | $62.12 \pm 1.73$ | $57.42 \pm 6.20$ | $6.94 \pm 6.58^{*}$ | $\mathbf{63.42 \pm 1.55}$ | $15.80 \pm 4.03^{*}$ |
|  | PE-Action (Ours) | $\mathbf{61.62 \pm 1.90}$ | $59.06 \pm 5.18$ | $3.80 \pm 3.60^{*}$ | $14.67 \pm 7.53^{*}$ | $53.83 \pm 2.64^{*}$ |
| Four Rooms | PE-Value (Ours) | $\mathbf{25.33 \pm 0.56}$ | $17.61 \pm 1.99^{*}$ | $17.27 \pm 2.42^{*}$ | $13.00 \pm 1.98^{*}$ | $24.05 \pm 0.84$ |
|  | PE-Action (Ours) | $\mathbf{22.98 \pm 0.76}$ | $12.61 \pm 3.11^{*}$ | $19.93 \pm 1.83$ | $21.18 \pm 1.02$ | $21.81 \pm 0.97$ |
| Fetch Reach | PE-Value (Ours) | $29.72 \pm 2.95$ | $16.40 \pm 3.50^{*}$ | $15.94 \pm 4.35^{*}$ | $23.51 \pm 3.54$ | $\mathbf{33.55 \pm 1.54}$ |
|  | PE-Action (Ours) | $\mathbf{34.76 \pm 1.94}$ | $18.08 \pm 3.71^{*}$ | $14.07 \pm 2.98^{*}$ | $23.37 \pm 2.78^{*}$ | $33.56 \pm 1.20$ |

Table 3: Ablation of teacher reward functions with fixed PE state representations. Reporting mean area under the student's learning curve plus/minus standard error. The results are over 10 runs. * Indicates a significant difference (p<.05) between our LP reward function and the baseline reward functions. Bold indicates the highest mean area under the curve.

We first compare both variants of our parametric-behavior embedder, PE-Values and PE-Actions, against the student parameters (Narvekar et al., 2017) and the heuristic state representation used by Fan et al. (2018). In this setting, the teacher's reward is fixed to be our LP reward. Overall, we found that the PE state representation is a more robust teacher state representation as the student's environments get more complex. With our PE state representations, the teacher's curriculum policy resulted in a higher AUC for the student in both Four Rooms and Fetch Reach environments (see Table 2).

Next, we compare our LP reward against the reward functions used in Narvekar et al. (2017), Fan et al. (2018), Ruiz et al. (2019) and Matiisen et al. (2020). In this setting, the teacher's state representation is fixed to be either our PE-Actions or PE-Values representation. We found that in 4/6 of our experiments (student environment x PE variant), the student achieves a higher AUC value when trained with a teacher utilizing the LP reward (see Table 3). Moreover, we found that in both the reward and state ablation experiments, by using our LP reward or PE state representations, the teacher has comparable or improved learning efficiency across the differing student environments (see Figures 19 and 20 in Appendix K). To that end, we have successfully demonstrated that (1) Reinforcement Teaching can be used to learn effective curricula that improve student learning and (2) our PE state representations and LP reward function are important elements of our framework.

---

[3]The Welch t-test was found to be more robust to violations of their assumptions compared to other parametric and non-parametric tests (e.g., t-test, ranked t-test) (Colas et al., 2019). In certain results we found the normality assumption to be violated, therefore the Welch t-test a better choice than others.

## 5.2 Step-size Adaptation for Supervised Learning Students

For the step-size adaption setting, the goal is for the teacher to learn a policy that adapts the step-size of a student's base optimizer. The student is a supervised learning algorithm with the objective of learning a synthetic classification task. See Appendix I.2 for more details on the classification task. Learning a step-size adaptation policy that improves over a tuned optimizer is a challenging problem because of the effectiveness of natively adaptive optimizers, such as Adam (Kingma & Ba, 2015).

**Teaching MDP for Step-size Adaption**  We start by formalizing the teaching MDP for the step-size adaption problem setting (see Table 1). For this problem, the teacher will control the step-size of the student's optimizer. More specifically, the teacher's action is a relative change in the step-size, doubling it, halving it, or remaining constant. For each step in the student's learning process, the student neural network takes a gradient step with a step-size determined by the teacher.

For the PE state representation, we fix the mini-state size at 256 and include three variations: PE-0, which observes only outputs, PE-X, which observes inputs and outputs, and PE-Y, which observes targets and outputs. In this setting, the inputs are the features $x_i$, the outputs are the classifier's predictions, $f_\theta(x_i)$, and the targets are the ground truth labels $y_i$. Lastly, for all reward functions, the performance measure is the student's validation accuracy.

**Teacher Training**  To train the teacher, we use a variant of DQN, Double DQN, and follow the pseudocode in Algorithm 1. As discussed in Section 5.1, we use DQN style algorithms for the teacher because of their simplicity and sample efficiency. Our use of Double DQN here also shows that our results are robust to different choices of RL algorithms. See Appendix J.2 for full details on teacher and student hyperparameters.

**Teacher Evaluation**  We evaluate the teacher in a similar manner as mentioned in Section 5.1. The teacher's policy is fixed and then evaluated on a newly initialized student. One goal is to determine whether the teacher learned an effective policy to adapt the student optimizer's step-size over time. Therefore, we show the student's learning curve, while the student uses the step-sizes proposed by the teacher (see Figures 5-right and 6-right). It is also important that our Reinforcement Teaching approach is sample-efficient, therefore we show the teacher's own learning curve during training (see Figures 5-left and 6-left). Furthermore, we perform a policy-transfer experiment, where we demonstrate that with our approach, the teacher can learn a step-size adaptation policy that can be transferred to new students classifying different benchmark datasets (MNIST, Fashion MNIST) and even new students with different architectures (see Figure 7).

To compare against existing work, we first conduct two ablation studies on the state representation using SGD and Adam as the base optimizers for the synthetic classification task. We compare the variants of our parametric-behavior embedder against (1) student parameters (Narvekar et al., 2017), (2) Policy Evaluation Networks (PVNs), and (3) a heuristic state representation that contains the time-step, train accuracy and validation accuracy. The heuristic state is representative of previous work like L2T (Fan et al., 2018).

Moreover, in the same synthetic classification task with the Adam optimizer, we further ablate the reward of Reinforcement Teaching, comparing our LP reward function to the time-to-threshold reward (Narvekar et al., 2017) and the L2T reward (Fan et al., 2018). All results are averaged over 30 random seeds, and the shaded regions are 95% CIs.

**Ablating State Representations**  Using SGD as the base optimizer, we use an easy synthetic classification task where most random teacher trajectories can reach the performance threshold. We do this to disentangle any effects of the reward function, and use only the time-to-threshold reward. We find that using the PE variants significantly increases the teacher's learning efficiency compared to the baselines (see Figure 5-top left). In particular, PE-X is slower to fit the data because it must learn to fit the Gaussian inputs, whereas PE-0 is able to more quickly learn from its smaller state representation (while this seems surprising that outputs alone are effective, we discuss why this is effective in supervised learning in Appendix E). Both the PVN and the parameter state representation are no better than the simple heuristic in this problem. Observing the learning rate schedule that the teaching policy induces in Figure 21, we see that the parameter state representation uses a nearly constant learning rate and is not adaptive. The parameter state

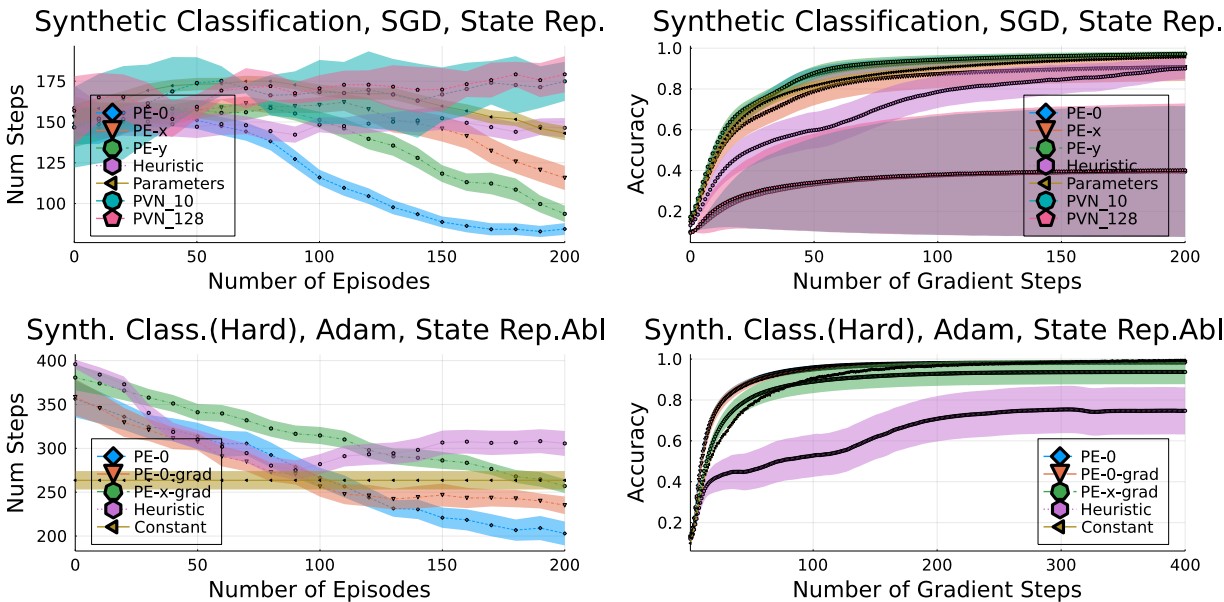

Figure 5: State ablation experiments. The left plots are learning curves for the teacher. The y-axis is the number of gradient steps needed for the student to reach the performance threshold with the teacher's current policy, as the teacher learns over episodes on the x-axis (lower is better). The right plots are the student's training curves while using the trained teacher's step-size policy (higher is better). Top: student's base optimizer is SGD. Bottom: student's base optimizer is Adam, classification task is harder.

representation is the Markov state for SGD, but, learning from parameters is difficult even for this student's 2-layer neural network (19k parameters). PVN is also unable to improve even after increasing the number of probing inputs from 10 to 128. Furthermore, with respect to the student's learning curve in Figure 5 (top right), we see similar results as previously found in the Curriculum Learning setting. With the PE state representations, the teacher is able to output a policy that either improves the student's learning efficiency or results in greater final validation accuracy compared to the baselines.

**Ablating Mini-state Size**   Using the same synthetic classification problem as before, with the student using the SGD optimizer, we now ablate PE's mini-state size (*i.e.* the number of inputs and outputs used before pooling). In Figure 6 (bottom center), we find that the teacher improves with larger mini-state sizes. However, even a mini-state size of 32 provides a state representation that is able to improve over the baselines: heuristic, parameters, and PVNs.

**Ablating State Representation With Adam as Base Optimizer**   We now conduct an experiment with Adam as the base optimizer and with a more difficult synthetic classification task. The only difference in this synthetic classification task, is that the performance threshold is higher. This is needed because Adam can quickly solve the previous synthetic classification task with a large range of constant learning rates. Adam uses a running trace of the parameter gradients in the momentum term, so the Reinforcement Teaching MDP is no longer Markov in the parameters. To account for momentum, the mini-state can be augmented to include, in addition to the inputs and outputs, the change in outputs after a gradient step (denoted by -grad in legend, see Appendix D for details). Referring to Figure 5 (bottom left and right), we find that PE is the *only* state representation to improve over Adam with the best constant step-size. More specifically, we found that in 200 teacher episodes, by using the PE state representations the teacher can learn a step-size adaption policy that results in the student reaching its performance threshold in approximately 200 time-steps. Given the same amount of teacher training time with the heuristic state, the teacher's policy only enables the student to reach its performance threshold after 300 time-steps (see Figure 5-bottom left). Moreover, it is surprising to note that PE-0 is the best performing state representation despite not being a

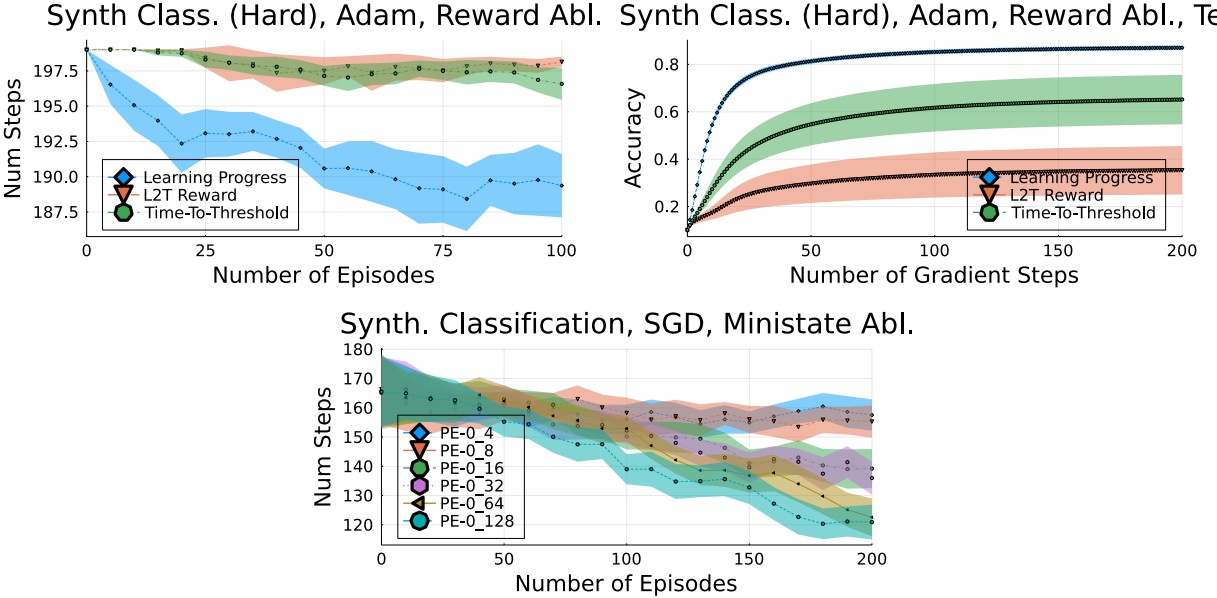

Figure 6: Top: Reward ablation experiments where the teacher adapts step-size of Adam. The left plot is the learning curve for the teacher (lower is better). The right plot is the student's training curves while using the trained teacher's step-size policy (higher is better). Bottom: Teacher's learning curve ablating the PE mini-state size where the teacher adapts the SGD optimizer's step size.

Markov state representation for this problem. The policy learned by Reinforcement Teaching with PE also successfully transfers to new architectures (see Appendix L.2).

**Ablating Reward Functions**    Using Adam, PE-0, and the hard synthetic classification problem from the previous experiment, we now ablate the reward function of Reinforcement Teaching. The earlier experiments were designed to be insensitive to the reward function in such a way that a random teaching policy would reach the performance threshold. We note that the policy found in the Adam experiments can reach the performance threshold in under 200 steps, while the initially random policy takes more than 350 steps. We now ablate reward shaping with a max steps of only 200, making both the L2T and time-to-threshold reward relatively sparse due to time-outs. Referring to Figure 6 (top-left), we find that learning progress shapes the reward and allows the teacher to learn a step-size adaptation policy that improves over Adam in only 100 teacher episodes, compared to 200 episodes in the previous experiment in Figure 5 (bottom-left). This indicates that our LP reward function is important for improving the teacher's learning efficiency. Similarly, we found that with the LP reward the teacher's policy significantly improves the student's final validation accuracy and learning efficiency compared to the other reward baselines (see Figure 6- top right).

**Transferring the Policy**    To learn a general step-size adaptation policy, which is effective across benchmark datasets, the teacher must train students on a large range of optimization problems. We now conduct experiments in which the teacher learns in the "Neural Network Training Gym" environment, in which we sample a new synthetic classification task at the beginning of each episode. The teacher then learns to adapt the step-size for the student's neural network on that classification task for that episode. While synthetic, this problem covers a large range of optimization problems by varying the classification task at each episode. After training the teacher's policy in the NN Training Gym, we transfer the policy to adapt the step-size for a student learning on benchmark datasets: MNIST (LeCun et al., 2010) and Fashion-MNIST (Xiao et al., 2017). This transfer experiment changes not only the data, but also the student's neural network architecture (see details in Appendix I.2). We find that the heuristic state representation is able to reach the performance threshold for the synthetic data (see Figure 7-top left). Referring to Figure 7 (top-right and bottom), the heuristic teaching policy does not transfer well to benchmark datasets. Our PE state representation, how-

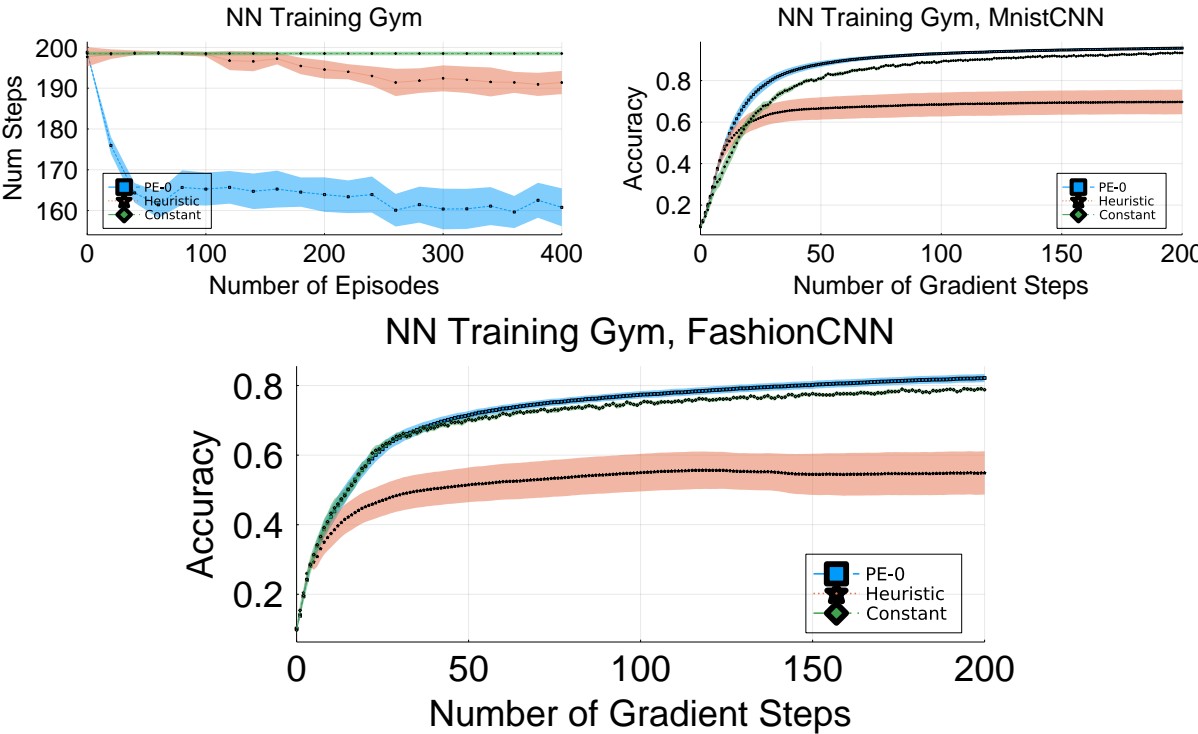

Figure 7: Reinforcement Teaching in the Neural Network Training Gym. Student learning curves use either the best constant step-size or a step-size adaptation policy that was transferred after being learned in the training gym. Top-left: Teacher learning curves, lower is better. Top-right: Student learning curves with CNN on MNIST. Bottom: Student learning curves with CNN on Fashion MNIST.

ever, is able to transfer the step-size adaptation policy to both MNIST and Fashion MNIST, as well as to a student that is learning with a Convolutional Neural Network (CNN). This is surprising because the NN Training Gym did not provide the teacher with any experience in training students with CNNs.

## 6  Discussion

Our experiments have focused on a narrow slice of Reinforcement Teaching: meta-learning curricula for a reinforcement learner and the step-size of an optimizer for a supervised learner. However, several other meta-learning problems can be formulated using Reinforcement Teaching, such as learning to explore.

The main limitation of Reinforcement Teaching is the limitation of current RL algorithms. In designing the reward function, we used an episodic formulation because RL algorithms currently struggle in the continuing setting. Another limitation of the RL approach is that the dimensionality of the teacher's action space cannot be too large, such as directly parameterizing an entire neural network. While we have developed the parametric-behavior embedder to learn indirectly from parameters, an important extension of Reinforcement Teaching would be to learn to represent actions in parameter space.

In this paper, we presented Reinforcement Teaching: a general formulation for meta-learning using RL. To facilitate learning in the teacher's MDP, we introduced the parametric-behavior embedder that learns a representation of the student's parameters from behavior. For credit assignment, we shaped the reward with learning progress. We demonstrated the generality of Reinforcement Teaching across several meta-learning problems in RL and supervised learning. While an RL approach to meta-learning has certain limitations, Reinforcement Teaching provides a unifying framework that will continue to scale as RL algorithms improve.

## 7 Acknowledgements

Part of this work was done in the Intelligent Robot Learning (IRL) Lab at the University of Alberta. The authors would like to thank Antonie Bodley, as well as all the anonymous reviewers for their constructive feedback. This research was supported, in part, by funding from the Canada CIFAR AI Chairs program, Alberta Machine Intelligence Institute (Amii), Compute Canada, Huawei, Mitacs, Alberta Innovates, and the Natural Sciences and Engineering Research Council (NSERC).

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

## A  Code for Experiments

The source code to run our experiments can be found in this anonymized dropbox link: https://www.dropbox.com/sh/hjkzzgctnqf6d8w/AAAYEycaDvPOeifz8FZbR3kLa?dl=0

## B  Teacher's Action Space

The diagram highlights how the choice of action space for the teacher enable the teacher to learn varied policies that can be applied across different domains

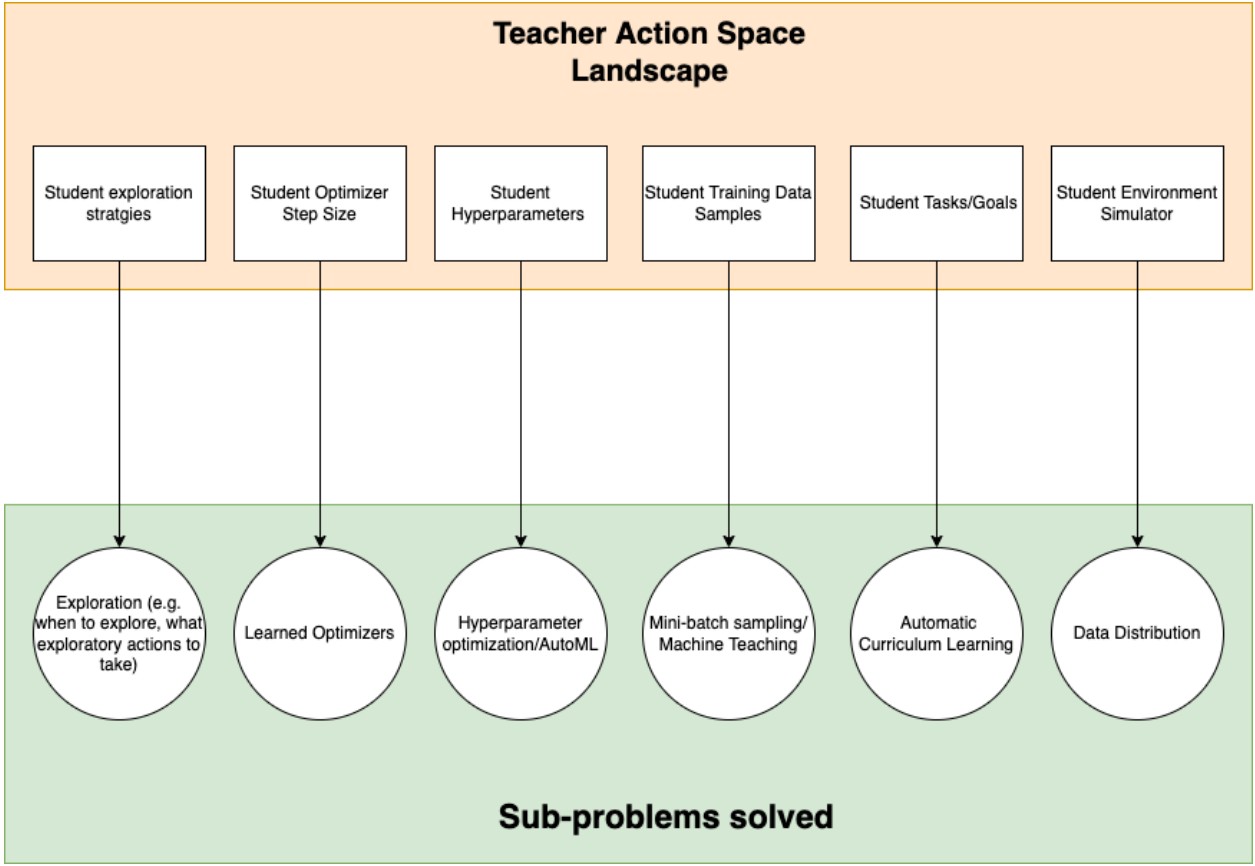

Figure 8

## C  More Details on Reward Functions

The reward function discussed in Section 4.3 is a time-to-threshold reward function for some threshold $m^*$. Another common criterion trains the learner for $T$ iterations and records the performance at the end. The learning process in this case is a fixed horizon, undiscounted, episodic learning problem and the reward is zero everywhere except that $r_T = m(\theta_T, \mathcal{D})$. In this setting, the policy that optimizes the learning progress also optimizes the final performance $m(\theta_T)$. Hence, adding learning progress can be seen as balancing the criteria previously discussed and in Section 4.3: reaching a performance threshold and maximizing overall performance.

For reward shaping, one issue with a linear potential is that a constant improvement in performance at lower performance levels is treated as equivalent to higher performance levels. Improving the performance of a classifier, for example, is much more difficult when the performance is higher. One way to account for this non-linearity in the classification setting is to introduce a non-linearity into the shaping, $\Phi(\theta) = \log(1 - m(\theta))$.

In the non-linear potential function, we may need to add $\epsilon$ to ensure numerical stability. With this nonlinear learning progress, the agent will receive higher rewards for increasing the performance measure at higher performance levels as opposed to lower ones.

In addition to learning progress, we can shape with only the new performance $m'$. Assuming that the performance measure is bounded, $0 \le m' \le 1$, such as for the accuracy of a classifier, we have that $-2 \ge -1 + m' \ge 0$. Because the reward function is still negative, it still encodes the time-to-threshold objective. This, however, changes the optimal policy. The optimal policy will maximize its discounted sum of the performance measure, which is analogous to the area under the curve.

When the performance measure $m$ is not bounded between 0 and 1, as is the case for the sum of rewards when the student is a reinforcement learner, we outline three alternatives. The first is to simply normalize the performance measure if a maximum and minimum is known. The second, when the maximum or minimum is not known, is to clip the shaping term to be between $-1$ and $1$. The last possibility, which is used when the scale of the performance measure changes such as in Atari (Mnih et al., 2015), is to treat any increase (resp. any decrease) in the performance measure as equivalent. In this case, we cannot use a potential function and instead shape with a constant, $F(s, a, s') = 2\,\mathbb{I}(\gamma m' - m > 0) - 1$. The teacher receives a reward of 1 for increasing the performance measure and a reward of $-1$ for decreasing the reward function. This also respects the structure of the time-to-threshold reward, while still providing limited feedback about the improvement in the agent's performance measure.

## D    Non-Markov Learning Settings

Most components of the learner's environment will not depend on more than the current parameters. Adaptive optimizers, however, accumulate gradients and hence depend on the history of parameters. In the context of reinforcement learning, this introduces partial observability. To enforce the Markov property in the teaching MDP, we would need to include the state of the optimizer or maintain a history of past states of the teaching MDP. Both appending the state of the optimizer and maintaining a history can be avoided by augmenting the mini-state $\hat{s} = \{x_i, f_\theta(x_i)\}_{i=1}^M$ with additional local information about the change due to a gradient step, $g_\theta(x_i) = f_{\theta - \alpha\nabla_\theta J}(x_i) - f_\theta(x_i)$ yielding $\hat{s}_{grad} = \{x_i, f_\theta(x_i), g_\theta(x_i)\}_{i=1}^M$. We will investigate the necessity of this additional state variable in Section 5.2.

## E    Learning From Outputs Alone in Stationary Problems

Each of the mini-states is a minibatch of inputs and outputs from the student. This means that training a teacher using stochastic gradient descent involves sampling a minibatch of minibatches. When the inputs are high-dimensional, such as the case of images, the mini-state that approximates the state can still be large. The inputs are semantically meaningful and provide context to the teacher for the outputs. Despite contextualizing the output value, the inputs put a large memory burden on training the teacher. We can further approximate the representation of the parameters by looking at the outputs alone.

To see this, suppose $h_{pool}$ is mean pooling and that the joint encoder $h_{joint}$ is a linear weighting of the concatenated input and output. Then the parametric behavior embedder simplifies $\frac{1}{M}\sum_{i=1}^M \mathbf{W}\big[x_i, f_\theta(x_i)\big] = \mathbf{W}\big[\frac{1}{M}\sum_{i=1}^M x_i, \frac{1}{M}\sum_{i=1}^M f_\theta(x_i)\big]$. For a large enough sample size, and under a stationary distribution $x \sim p(x)$, $\frac{1}{M}\sum_i x_i \approx \mathbb{E}[x_i]$ is a constant. Hence, if the minibatch batch size is large enough and the distribution on inputs is stationary, such as in supervised learning, we can approximate the state $\theta$ by the outputs of $f_\theta$ alone. While this intuition is for mean pooling and a linear joint encoding, we will verify empirically that this simplification assumption is valid for both a non-linear encoder and non-linear pooling operation in Section 5.2.

# F  Parametric-behavior Embedder As Approximating A Markov State Representation

If $\theta$ is a Markov state representation, then the Parametric-behavior Embedding (PE) of $\theta$ is an approximate Markov state representation. Requiring that $\theta$ is a Markov state representation is described in the paper and holds for SGD without momentum in supervised learning for example.

Denote the student's objective function as $J(\theta) = \mathbb{E}_{x,y\sim p(x,y)} J(f_\theta(x), y)$. We show how PE is approximately Markov by showing that the mini-state (which is the input to PE) can represent $J(\theta)$ (reward) and $\nabla_\theta J(\theta)$ (state-transition).

Note that for any particular $\theta$, the objective function is determined by the input $(x)$, output of the student $(f_\theta(x))$, and the target $(y)$. Hence, $J(\theta)$ is representable as function of the mini-state (the input to PE). It remains to show that the objective function after a step of gradient descent is also representable in terms of the mini-state.

Using a first-order Taylor expansion: $J(\theta'(\theta)) = J(\theta) + (\theta'(\theta) - \theta)\nabla_\theta J(\theta) + o(\alpha^2)$. As argued previously, $J(\theta) = \mathbb{E}_{x,y\sim p(x,y)} J(f_\theta(x), y)$ can be represented in terms of the mini-state. We now turn our attention to $(\theta'(\theta) - \theta)\nabla_\theta J(\theta) = \alpha\nabla_\theta J(\theta) \cdot \nabla_\theta J(\theta)$.

For a linear function, $f_\theta(x) = \theta x$, we have that $\nabla_\theta J(\theta) = \mathbb{E}_{x,y\sim p(x,y)} \left[ \nabla_{f_\theta(x)} J(f_\theta(x), y) x^\top \right]$. This means that $\nabla_\theta J(\theta)$ is also representable in terms of the mini-state, with inputs $x$, outputs $y$ and targets $f_\theta(x)$.

For deep linear networks and non-linear networks, we require that the mini-state includes the outputs of each layer. However, we have demonstrated empirically that we are able to learn policies using PE without this theoretically needed information in the mini-state. Our experiments show that even for deep non-linear neural networks, the student's outputs $(f_\theta(x))$ and the targets $(y)$ is enough to learn a policy that outperforms the Markov state $(\theta)$ and cruder heuristic approximations used in the literature.

# G  Efficiently Learning to Reinforcement Teach

One criterion for a good Reinforcement Teaching algorithm is low sample complexity. Interacting with the teacher's MDP and evaluating a teacher can be expensive, due to the student, its algorithm or its environment. A teacher's episode corresponds to an entire training trajectory for the student. Hence, generating numerous teacher episodes involves training numerous students. The teacher agent cannot afford an inordinate amount of interaction with the student. One way to meet the sample complexity needs of the teacher is to use off-policy learning, such as Q-learning. Offline learning can also circumvent the costly interaction protocol, but may not provide enough feedback on the teacher's learned policy. There is a large and growing literature on offline and off-policy RL algorithms (Yu et al., 2020; Wu et al., 2019; Fujimoto & Gu, 2021; Kumar et al., 2020). However, we found that DQN (Mnih et al., 2015; Riedmiller, 2005) and DoubleDQN (van Hasselt, 2010; Van Hasselt et al., 2016) were sufficient to learn adaptive teaching behaviour and leave investigation of more advanced deep RL algorithms for future work.

# H   Connecting Reinforcement Teaching to Gradient-Based Meta-Learning

Summarized briefly, gradient-based meta-learning (or meta-gradient) methods learn some traditionally non-learnable parts of a machine learning algorithm by backpropagating through the gradient-descent learning update. Meta-gradient's broad applicability, relative simplicity, and overall effectiveness make it a common framework for meta-learning. For example, Model-Agnostic Meta Learning (MAML) is a meta-gradient method that can be applied to any learning algorithm that uses gradient-descent to improve few-shot performance (Finn et al., 2017) and similar ideas have been extended to continual learning (Javed & White, 2019) and meta RL (Xu et al., 2018). Here we outline how MAML and other meta-gradient methods can be viewed relative to Reinforcement Teaching.

When $\mathcal{A}lg$ and $m$ are both differentiable, such as when $\mathcal{A}lg$ is an SGD update on a fixed dataset, meta-gradient methods unroll the computation graph to optimize the meta objective directly. Using MAML as an example, the meta-gradient $\frac{\partial}{\partial \theta_0} m(f_{\theta_T}, \mathcal{D})$ can be compute by noticing that $f_{\theta_T} = \mathcal{A}lg(f_{\theta_{t-1}}, \mathcal{D})$ and expanding recursively, we have

$$m(f_{\theta_T}, \mathcal{D}) = m(\mathcal{A}lg(f_{\theta_{T-1}}, \mathcal{D}), \mathcal{D}) = m(\mathcal{A}lg(\cdots \mathcal{A}lg(f_{\theta_0}, \mathcal{D})), \mathcal{D}).$$

Using the language of Reinforcement Teaching, we can express MAML and other meta-gradient algorithms as a type of Reinforcement Learning algorithm. Meta-gradient algorithms make use of the known gradient-update model and its connection to the teaching MDP's state-transition model. If the teacher is interacting with the teaching MDP in such a way that the state-transition model is only a gradient update, then a meta-gradient algorithm is analogous to a type of model-based trajectory optimization method. In trajectory optimization, an action sequence is planned by unrolling both the state-transition model and the reward model in simulation. Then, the first action in the sequence is taken by the teacher. At the next time step, the teacher must execute the planning procedure again. This planning procedure is costly, requiring memory proportional to the product of the state-space size and the planning horizon, $O(|S||T|)$. Others have also noted, that meta-gradient learning can have difficult to optimize loss landscapes especially as the unrolling length of the computation graph increases (Flennerhag et al., 2022). We remark, however, that the Reinforcement Teaching approach described in this work is not mutually exclusive to meta-gradient methods. An interesting direction for future work is combining both model-free (Reinforcement Teaching) and model-based (meta-gradient) in one meta-learning algorithm.

Lastly, we provide further discussion on why gradient-based meta-learning is not a good approach for the problems that we address in our experiments. Gradient-based meta-learning, which can be thought of as model-based trajectory optimization, is only applicable if 1) we have the state-transition model, 2) that state-transition model is fully-differentiable and 3) planning an entire trajectory using the fully-differentiable model is computationally feasible. In our reinforcement learning experiments, the state-transition model is not just a gradient-update but a sequence of interactions between the student's policy and its environment. While the gradient-update itself is differentiable, the interaction between the student's policy and the environment is not differentiable. Hence, the state-transition model is not differentiable, making gradient-based meta-learning inapplicable. For our supervised learning experiments, gradient-based meta-learning is applicable in principle because the state-transition model is just a gradient-update. As discussed in Section 2, however, gradient-based optimization requires extensive trajectory-based optimization that is specific to each state and student or architecture. Reinforcement teaching allows us to learn a policy that can be queried quickly at any step and for any student or architecture, but gradient-based meta-learning requires expensive re-computation at each time-step. Hence gradient-based meta-learning is computationally infeasible, and less generalizable to changes in the meta-learning problem, compared to Reinforcement Teaching.

# I Environment and Baseline Specification

In this section, we will outline the environments used for both the RL and supervised learning experiments.

## I.1 Environments for RL experiments

**Maze**   The Maze environment is an $11 \times 16$ discrete grid with several blocked states (see Figure 9). An agent can take four deterministic actions: up, down, left, or right. If an agent's action takes the agent off the grid or into a blocked state, the agent will remain in its original location. See Table 4 for details on the environment reward. To make this environment more difficult, we limited the max number of time-steps per episode to only 40. Therefore, the agent cannot simply randomly explore until it reaches the goal. Furthermore, in this environment, the teacher's action will change the student's start state. The teacher can start the student at 11 possible locations, including the start state of the target task. The teacher's action set contains both impossible tasks (e.g., start states that are completely blocked off) and irrelevant tasks (e.g., start states that are not necessary to learn for the target task). This environment is useful to study for several reasons. First, the reduced maximum time-step makes exploration difficult thus curriculum learning becomes a necessity. Secondly, the set of impossible and irrelevant sub-tasks in the teacher's action set ensure that the teacher is able to learn to avoid these actions and only suggest actions that enable the student to learn the target task efficiently (i.e., navigating from the blue to green state, see Figure 9).

**Four Rooms**   The Four Rooms environment is adapted from the MiniGrid suite Chevalier-Boisvert et al. (2018). It is a discrete state and action grid-world. Although the state space is discrete, it is very large. The state encodes each grid tile with a 3 element tuple. The tuple contains information on the color and object type in the tile. Due to the large state space, this environment requires a neural network function approximator on behalf of the RL student agent. The large state space makes Four Rooms much more difficult than the tabular Maze environment. Similar to the Maze domain, Four Rooms has a fixed start and goal state, as shown in see Figure 10. In addition, the objective is for an agent to navigate from the start state to the goal state as quickly as possible. In our implementation, we used the compact state representation and reward function provided by the developers. The state representation is fully observable and encodes the color and objects of each tile in the grid. See Table 4 for more details on the environment.

**Fetch Reach**   Fetch Reach is a continuous state and action simulated robotic environment Plappert et al. (2018). It is based on a 7-DoF Fetch robotics arm, which has a two-fingered parallel end-effector (see Figure 11). In Fetch Reach, the end-effector starts at a fixed initial position, and the objective is to move the end-effector to a specific goal position. The goal position is 3-dimensional and is randomly selected for every episode. Therefore, an agent has to learn how to move the end-effector to random locations in 3D space. Furthermore, the observations in this environment are 10-dimensional and include the Cartesian position and linear velocity of the end-effector. The actions are 3-dimensional and specify the desired end-effector movement in Cartesian coordinates. See Table 4 for more details on the environment. The teacher controls the goal distribution. The goal distribution determines the location the goal is randomly sampled from. There are 9 actions in total, each action gradually increasing the maximum distance between the goal distribution and the starting configuration of the end-effector. Therefore, "easier" tasks are ones in which the set of goals are very close to the starting configuration. Conversely, "harder" tasks are ones in which the set of goals are far from the starting configuration of the end-effector. It is important to note, however, that the goal distribution of each action subsumes the goal distribution of the previous action. For example, if action 1 allows the goal to be sampled within the interval $[0, .1]$, then action 2 allows the goal to be sampled within the interval $[0, .2]$. This allows for learning on "easy" tasks to be useful for learning on "harder" tasks.

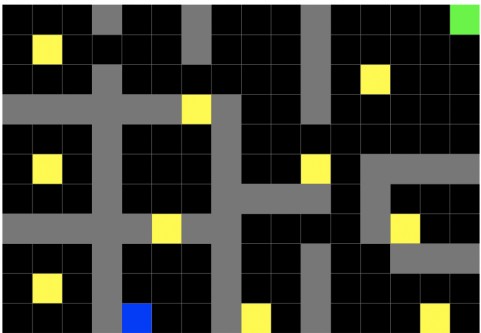

Figure 9: Maze: The green square represents the goal state, and the blue square represents the start state of the target task. Yellow squares indicate the teacher's possible actions — possible starting states for the student.

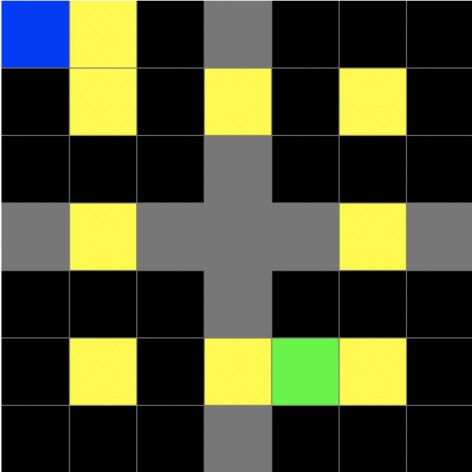

Figure 10: Four Rooms: The green square represents the goal state, and the blue square represents the start state of the target task. Yellow squares indicate the teacher's possible actions — possible starting states for the student.

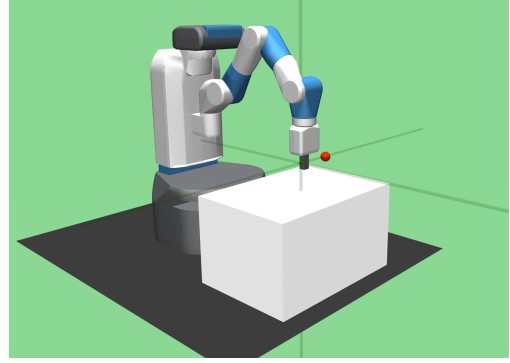

Figure 11: Fetch Reach

| | Maze | Four Rooms | Fetch Reach |
|---|---|---|---|
| **Env action type** | Discrete | Discrete | Continuous |
| **Number of env actions** | 4 | 3 | NA |
| **Env state space type** | Discrete | Continuous | Continuous |
| **Dimension of env state** | 1 | 243 | 10 |
| **Max number of time-steps** | 40 | 40 | 50 |
| **Env reward** | $R(t) = 0$ except $R(T) = (.99)^T$ | $R(t) = 0$ except $R(T) = 1 - 0.9 * \frac{T}{maxsteps}$ | $R(t) = -1$ except $R(T) = 0$ |
| **Performance Threshold** | .77 (discounted return) | .6 (discounted return) | .9 (success rate) |

Table 4: Environment characteristics. T denotes the time-step at termination.

**RL Experiment Baselines** For the L2T Fan et al. (2018) baseline, we used the reward function exactly as described in the paper. For the state representation, we used an approximation of their state which consisted of the teacher's action, the student's target task score, the source task score, and the student episode number. For the Narvekar et al. (2017) baseline, we used the time-to-threshold reward function which is a variant of their reward function. For the state, we used the student parameters, as described in their paper. Lastly, for the Matiisen et al. (2020) baseline, we implemented it as directed by the pseudocode in the paper. We also swept over the tau and alpha hyperparameters, as those were the only hyperparameters required. For both, we swept over the values in $\{.01, .1, .5, 1.0\}$.

## I.2 Supervised Learning

We describe the classification datasets used by the student. Note that the teacher's action is a relative change in the step size, so we also append the current step-size for all state representations.

**Synthetic Classification**: At the beginning of each episode, we initialize a student neural network with 2 hidden layers, 128 neurons, and relu activations. The batch size is 64. For each episode, we also sample data $x_i \sim N(\mathbf{0}, \mathbf{I})$, $i = 1, \ldots, 1000$ and $\mathbf{0} \in \mathbb{R}^{10}$ and $\mathbf{I}$ is the identity matrix. Each $x_i$ is labelled $y_i \in 1, \ldots, 10$ according to its argmax $y_i = \arg\max x_i$. For each step in the environment, the student neural network takes a gradient step with a step size determined by the teacher. We use a relative action set, where the step size can be increased, kept constant or decreased. This problem was designed so that the default step size of the base optimizer would be able to reach the termination condition within the 200 time steps allotted in the episode. Exploration is not a requirement to solve this problem, as we are primarily evaluating the state representations for Reinforcement Teaching and the quality of the resulting policy.

- SGD Variant: Termination condition based on performance threshold of $m^* = 0.95$, max steps is 200.

- Adam Hard Variant: Termination condition based on performance threshold of $m^* = 0.99$, max steps is 400.

**Neural Network Training Gym**: At the beginning of each episode, we initialize a student neural network with 2 hidden layers, 128 neurons, and relu activations. The batch size is 128. For each episode, we also sample data $x_i \sim N(\mathbf{0}, \mathbf{I})$, $i = 1, \ldots, 4000$ and $\mathbf{0} \in \mathbb{R}^{784}$ and $\mathbf{I}$ is the identity matrix. The data $x_i$ are classified by a randomly initialized labeling neural network $y_i = f^*(x_i)$. The labeling neural network $f^*$ has the same number of layers as the student's neural network but has 512 neurons per layer and tanh activations to encourage a roughly uniform distribution over the 10 class labels.

**MNIST**: The student's neural network is a feed-forward neural network with 128 neurons and 2 hidden layers. The CNN Variant uses a LeNet5 CNN with a batch size of 64. Subsampled dataset to 10000 so that an episode covers one epoch of training.
**Fashion-MNIST**: The student's neural network is a feed-forward neural network with 128 neurons and 2 hidden layers. The CNN Variant uses a LeNet5 CNN with a batch size of 256. Subsampled dataset to 10000 so that an episode covers one epoch of training.

**CIFAR-10**: The student's neural network is a LeNet5 CNN wih a batch size of 128. Subsampled dataset to 10000 so that an episode covers one epoch of training.

## J   Hyperparameters for Experiments

In this section, we will outline all hyperparameters used for the RL and supervised learning experiments.

### J.1   Reinforcement Learning Experiments

**Teacher Hyperparameters**   In the Maze experiments, for the DQN teacher, we performed a grid search over batch size $\in \{64, 128, 256\}$, learning rate $\in \{.001, .005\}$, and minibatch $\in \{75, 100\}$. Next, in the Four Rooms experiments, for the DQN teacher, we performed a grid search over batch size $\in \{128, 256\}$, and minibatch $\in \{75, 100\}$. We use a constant learning rate of .001. Lastly, in the Fetch Reach experiments, for the DQN teacher, we performed a grid search over batch size $\in \{128, 256\}$. We use a constant learning rate of .001 and mini-batch size of 200. The best hyperparameters for each of the baselines are reported in the tables.

| | Hyperparameters used across all envs |
|---|---|
| **Teacher agent type** | DQN |
| **Optimizer** | ADAM |
| **Gamma** | .99 |
| **Tau** | $10^{-3}$ |
| **Target network update frequency** | 1 |
| **Starting epsilon** | .5 |
| **Epsilon decay rate** | .99 |
| **Value network** | 2 layers with 128 units each, Relu activation |

Table 5: Fixed teacher hyperparameters used across all methods.

| Maze | | |
|---|---|---|
| **Baseline** | **Batch size** | **Learning rate** |
| L2T state and LP reward | 128 | .001 |
| Student parameters state and LP reward | 64 | .001 |
| **Four Rooms** | | |
| **Baseline** | **Batch size** | **Learning rate** |
| L2T state and LP reward | 128 | .001 |
| Student parameters state and LP reward | 128 | .001 |
| **Fetch Reach** | | |
| **Baseline** | **Batch size** | **Learning rate** |
| L2T state and LP reward | 256 | .001 |
| Student parameters state and LP reward | 128 | .001 |

Table 8: Teacher agent hyperparameters for teacher state ablation experiments.

| Maze | | | | | |
|------|------|------|------|------|------|
| **Baseline** | **Batch size** | **Learning rate** | **Mini-batch size** | **Tau** | **Alpha** |
| PE-Actions and LP (Ours) | 256 | .001 | 100 | NA | NA |
| PE-Values and LP (Ours) | 256 | .005 | 100 | NA | NA |
| Narvekar et al. (2017) | 64 | .001 | NA | NA | NA |
| L2T Fan et al. (2018) | 128 | .005 | NA | NA | NA |
| TCSL Online | NA | NA | NA | 0.1 | 1.0 |
| **Four Rooms** | | | | | |
| **Baseline** | **Batch size** | **Learning rate** | **Mini-batch size** | **Tau** | **Alpha** |
| PE-Actions and LP (Ours) | 128 | .001 | 100 | NA | NA |
| PE-Values and LP (Ours) | 128 | .001 | 100 | NA | NA |
| Narvekar et al. (2017) | 256 | .001 | NA | NA | NA |
| L2T Fan et al. (2018) | 128 | .001 | NA | NA | NA |
| TCSL Online | NA | NA | NA | 0.1 | 1.0 |
| **Fetch Reach** | | | | | |
| **Baseline** | **Batch size** | **Learning rate** | **Mini-batch size** | **Tau** | **Alpha** |
| PE-Actions and LP (Ours) | 256 | .001 | 200 | NA | NA |
| PE-Values and LP (Ours) | 256 | .001 | 200 | NA | NA |
| Narvekar et al. (2017) | 256 | .001 | NA | NA | NA |
| L2T Fan et al. (2018) | 128 | .001 | NA | NA | NA |
| TCSL Online | NA | NA | NA | 0.1 | 0.5 |

Table 6: Teacher agent hyperparameters for all methods (excluding ablation experiments).

**Teacher-Student Protocol Hyperparameters**   This section contains information about the hyperparameters used for the teacher-student interaction protocol.

| | Maze | Four Rooms | Fetch Reach |
|------|------|------|------|
| **Student training iterations** | 100 | 50 | 50 |
| **# episodes/epochs per student training iteration** | 10 | 25 | 6 |
| **# evaluation episodes/epochs per student training iteration** | 30 | 40 | 80 |
| **# of teacher episodes** | 300 | 100 | 50 |

Table 9: Hyperparameters used in the teacher-student training procedure.

**Student Hyperparameters**   For the PPO student, we used the open-source implementation in (Willems & Karra, 2020). For the DDPG student, we used the OpenAI Baselines implementation Dhariwal et al. (2017). We used the existing hyperparameters as in the respective implementations. We did not perform a grid search over the student hyperparameters.

| Maze | | | |
|---|---|---|---|
| **Baseline** | **Batch size** | **Learning rate** | **Mini-batch size** |
| PE-Actions and Time-to-threshold | 256 | .001 | 100 |
| PE-Values and Time-to-threshold | 128 | .005 | 75 |
| PE-Actions and L2T reward | 256 | .001 | 75 |
| PE-Values and L2T reward | 256 | .001 | 100 |
| PE-Actions and Ruiz et al. (2019) reward | 128 | .001 | 100 |
| PE-Values and Ruiz et al. (2019) reward | 64 | .001 | 100 |
| PE-Actions and Matiisen et al. (2020) reward | 128 | .001 | 75 |
| PE-Values and Matiisen et al. (2020) reward | 128 | .001 | 75 |
| **Four Rooms** | | | |
| **Baseline** | **Batch size** | **Learning rate** | **Mini-batch size** |
| PE-Actions and Time-to-threshold | 256 | .001 | 75 |
| PE-Values and Time-to-threshold | 256 | .001 | 100 |
| PE-Actions and L2T reward | 256 | .001 | 75 |
| PE-Values and L2T reward | 256 | .001 | 100 |
| PE-Actions and Ruiz et al. (2019) reward | 256 | .001 | 75 |
| PE-Values and Ruiz et al. (2019) reward | 128 | .001 | 100 |
| PE-Actions and Matiisen et al. (2020) reward | 256 | .001 | 75 |
| PE-Values and Matiisen et al. (2020) reward | 128 | .001 | 75 |
| **Fetch Reach** | | | |
| **Baseline** | **Batch size** | **Learning rate** | **Mini-batch size** |
| PE-Actions and Time-to-threshold | 256 | .001 | 200 |
| PE-Values and Time-to-threshold | 256 | .001 | 200 |
| PE-Actions and L2T reward | 256 | .001 | 200 |
| PE-Values and L2T reward | 128 | .001 | 200 |
| PE-Actions and Ruiz et al. (2019) reward | 128 | .001 | 200 |
| PE-Values and Ruiz et al. (2019) reward | 256 | .001 | 200 |
| PE-Actions and Matiisen et al. (2020) reward | 128 | .001 | 200 |
| PE-Values and Matiisen et al. (2020) reward | 256 | .001 | 200 |

Table 7: Teacher agent hyperparameters for teacher reward ablation experiments.

| | Maze | Four Rooms | Fetch Reach |
|---|---|---|---|
| **Student Agent Type** | Tabular Q Learning | PPO | DDPG |
| **Optimizer** | NA | ADAM | ADAM |
| **Batch size** | NA | 256 | 256 |
| **Learning rate** | .5 | .001 | .001 |
| **Gamma** | .99 | .99 | NA |
| **Entropy coefficient/Epsilon** | .01 | .01 | NA |
| **Adam epsilon** | NA | $10^{-8}$ | $10^{-3}$ |
| **Clipping epsilon** | NA | .2 | NA |
| **Maximum gradient norm** | NA | .5 | NA |
| **GAE** | NA | .95 | NA |
| **Value loss coefficient** | NA | .5 | NA |
| **Polyak-averaging coefficient** | NA | NA | .95 |
| **Action L2 norm coefficient** | NA | NA | 1 |
| **Scale of additive Gaussian noise** | NA | NA | .2 |
| **Probability of HER experience replay** | NA | NA | NA |
| **Actor Network** | NA | 3 layers with 64 units each, Tanh activation | 3 layers with 256 units each, ReLU activation |
| **Critic Network** | NA | 3 layers with 64 units each, Tanh activation | 3 layers with 256 units each, ReLU activation |

Table 10: Student hyperparameters.

### J.2 Supervised Learning Experiments

The teacher in the supervised learning experiment used DoubleDQN with $\epsilon$-greedy exploration and an $\epsilon$ value of 0.01. The batch size and hidden neural network size was 256. The action-value network had 1 hidden layer, but the state encoder has 2 hidden layers. There are three actions, one of which keeps the step size the same and the other two increase or decrease the step size by a factor of 2.

|  | Optenv Sgd | Optenv Adam | Optenv Miniabl |
|---|---|---|---|
| Init Num Episodes | 200 | 200 | 200 |
| Optimizer | ADAM | ADAM | ADAM |
| Batch Size | 256 | 256 | 256 |
| Update Freq | 100 | 100 | 100 |
| AgentType | DoubleDQN | DoubleDQN | DoubleDQN |
| Num Episodes | 200 | 200 | 200 |
| Num Env Steps | 2 | 2 | 2 |
| Hidden Size | 256 | 256 | 256 |
| Max Num Episodes | 200 | 200 | 200 |
| Activation | Relu | Relu | Relu |
| Num Grad Steps | 1 | 1 | 1 |
| Num Layers | 1 | 1 | 1 |
| Init Policy | Random | Random | Random |
| Gamma | 0.99 | 0.99 | 0.99 |
| Max Episode Length | 200 | 400 | 200 |

Table 11: Fixed hyperparameter settings for (Left-Right): SGD state ablation experiment, Adam state ablation experiment, Ministate ablation experiment.

|  | Optenv Reward | Optenv Pooling | Optenv Transfer |
|---|---|---|---|
| Init Num Episodes | 200 | 200 | 200 |
| Optimizer | ADAM | ADAM | ADAM |
| Batch Size | 256 | 256 | 256 |
| Update Freq | 100 | 100 | 100 |
| AgentType | DoubleDQN | DoubleDQN | DoubleDQN |
| Num Episodes | 400 | 400 | 400 |
| Num Env Steps | 2 | 2 | 2 |
| Hidden Size | 256 | 256 | 256 |
| Max Num Episodes | 200 | 200 | 200 |
| Activation | Relu | Relu | Relu |
| Num Grad Steps | 1 | 1 | 1 |
| Num Layers | 1 | 1 | 1 |
| Init Policy | Random | Random | Random |
| Gamma | 0.99 | 0.99 | 0.99 |
| Max Episode Length | 200 | 200 | 200 |

Figure 12: Fixed hyperparameter settings for (Left-Right): Reward shaping ablation experiment, Pooling Function ablation experiment, Transferring to real data experiment.

| Optenv Sgd | |
|---|---|
| Pooling Func | ["mean"] |
| Lr | [0.001, 0.0005, 0.0001] |
| State Representation | ["PE-0", "PE-x", "PE-y", "heuristic", "parameters", "PVN_10", "PVN_128"] |
| Num. Seeds | 30 |
| EnvType | ["OptEnv-NoLP-syntheticClassification-SGD"] |

Figure 13: Other specification and hyperparameters that are swept over in the SGD state ablation experiment.

| Optenv Adam | |
|---|---|
| Pooling Func | ["mean"] |
| Lr | [0.001, 0.0005, 0.0001] |
| State Representation | ["PE-0", "PE-0-grad", "PE-x-grad", "heuristic"] |
| Num. Seeds | 30 |
| EnvType | ["OptEnv-NoLP-syntheticClassification-ADAM"] |

Figure 14: Other specification and hyperparameters that are swept over in the Adam state ablation experiment.

| Optenv Miniabl | |
|---|---|
| Pooling Func | ["mean"] |
| Lr | [0.001, 0.0005, 0.0001] |
| State Representation | ["PE-0_4", "PE-0_8", "PE-0_16", "PE-0_32", "PE-0_64", "PE-0_128"] |
| Num. Seeds | 30 |
| EnvType | ["OptEnv-NoLP-syntheticClassification-SGD"] |

Figure 15: Other specification and hyperparameters that are swept over in the ministate size ablation experiment.

| Optenv Reward | |
|---|---|
| Pooling Func | ["mean"] |
| Lr | [0.001, 0.0005, 0.0001] |
| State Representation | ["PE-0"] |
| Num. Seeds | 30 |
| EnvType | OptEnv-["L2T","LP", "NoLP"]-syntheticClassification-ADAM |

Figure 16: Other specification and hyperparameters that are swept over in the reward ablation experiment.

| Optenv Pooling | |
|---|---|
| Pooling Func | ["attention", "max", "mean"] |
| Lr | [0.001, 0.0005, 0.0001] |
| State Representation | ["PE-0"] |
| Num. Seeds | 30 |
| EnvType | ["OptEnv-LP-syntheticClassification-ADAM"] |

Figure 17: Other specification and hyperparameters that are swept over in the pooling ablation experiment.

|  | Optenv Transfer |
|---|---|
| Pooling Func | ["mean"] |
| Lr | [0.001, 0.0005, 0.0001] |
| State Representation | ["PE-0", "heuristic"] |
| Num. Seeds | 30 |
| EnvType | ["OptEnv-LP-syntheticNN-ADAM"] |

Figure 18: Other specification and hyperparameters that are swept over in transferring to benchmark datasets experiment.

## K  Additional RL Experimental Results

A typical consequence of using RL to train a teacher is the additional training computation. In our method, there is both an inner RL training loop to train the student, and an outer RL training loop to train the teacher. Although this is true, we show that our method can greatly improve the teacher's learning efficiency and therefore reduce the overall amount of computation.

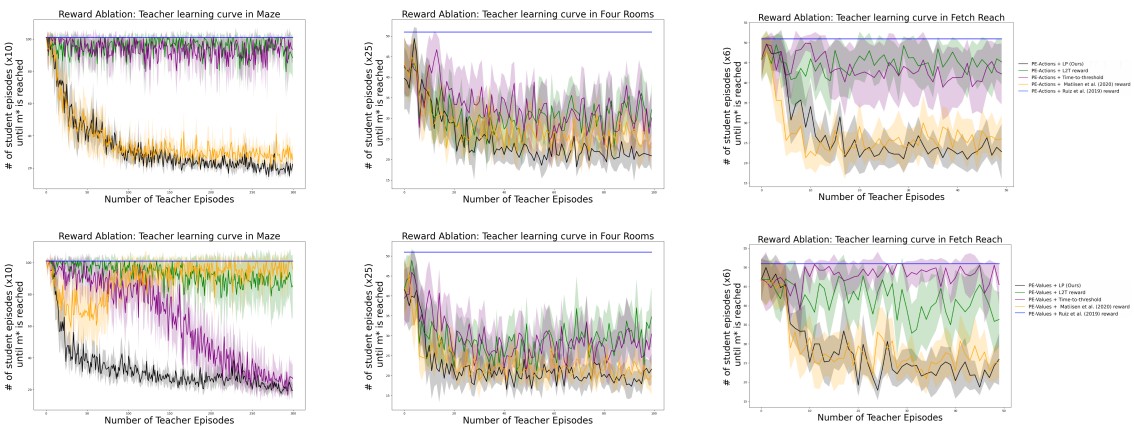

Figure 19: Left: Maze, Middle: Four Rooms, Right: Fetch Reach. This figure shows the teacher's training curves in the reward ablation experiments. Top: The state is fixed to be PE-Actions. Bottom: The state is fixed to be PE-Values. Lower is better.

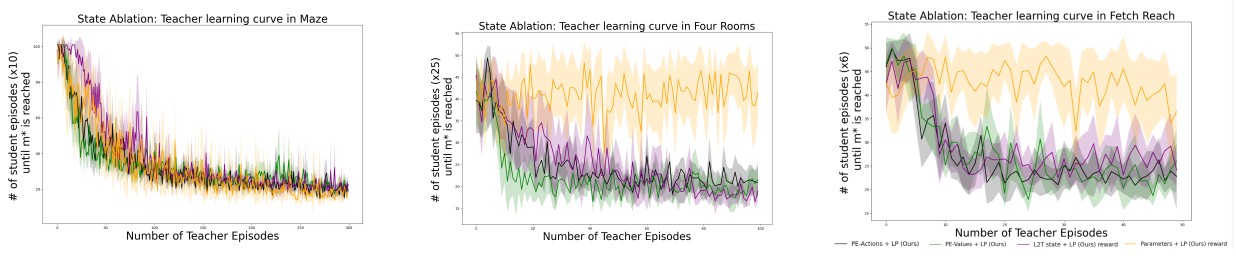

Figure 20: Left: Maze, Middle: Four Rooms, Right: Fetch Reach. This figure shows the teacher's training curves in the state ablation experiments. The reward is fixed to be LP. Lower is better.

# L    Additional Supervised Learning Experimental Results

## L.1    Training Curves with Base SGD Optimizer After Meta-Training

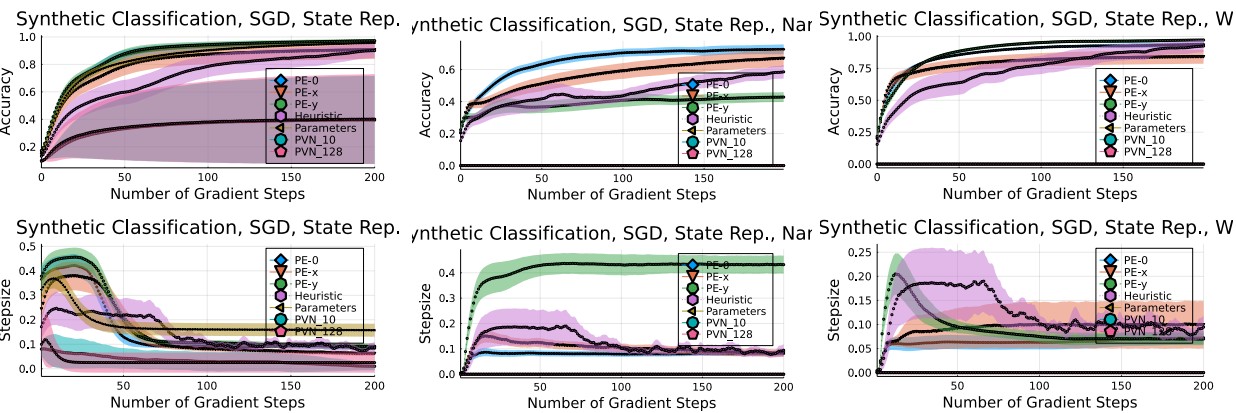

Figure 21: SGD State Ablation experiment. Top Student training curves with a trained teacher. Top: Step sizes selected by the teacher. Right: Same architecture as training. Center: A narrower but deeper architecture. Right: A wider but shallower architecture.

## L.2    Training Curves with Base Adam Optimizer After Meta-Training

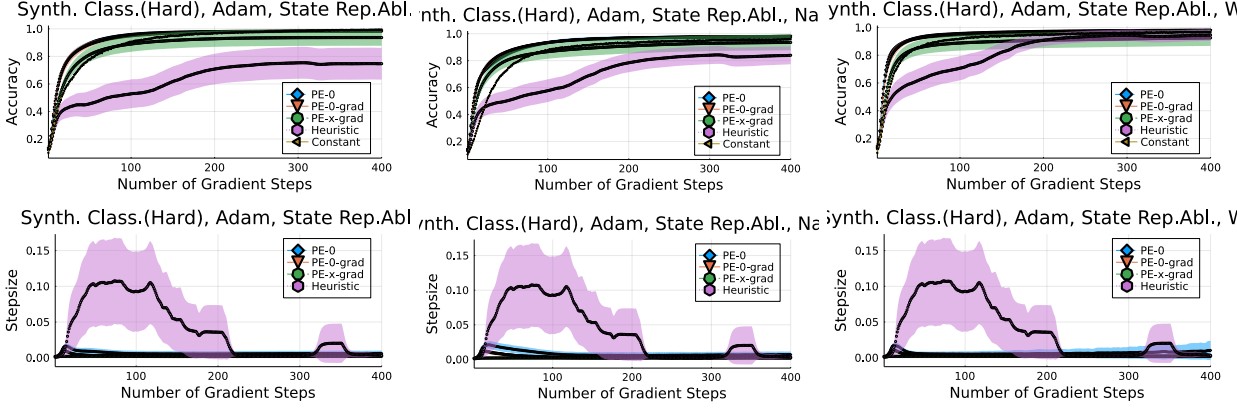

Figure 22: Adam State Ablation experiment. Top Student training curves with a trained teacher. Top: Step sizes selected by the teacher. Right: Same architecture as training. Center: A narrower but deeper architecture. Right: A wider but shallower architecture. Unlike using SGD as the base optimizer, the Reinforcement Teaching Adam optimizer generalizes well in both narrow and wide settings.

### L.3 Training Curves from Ministate Size Ablation

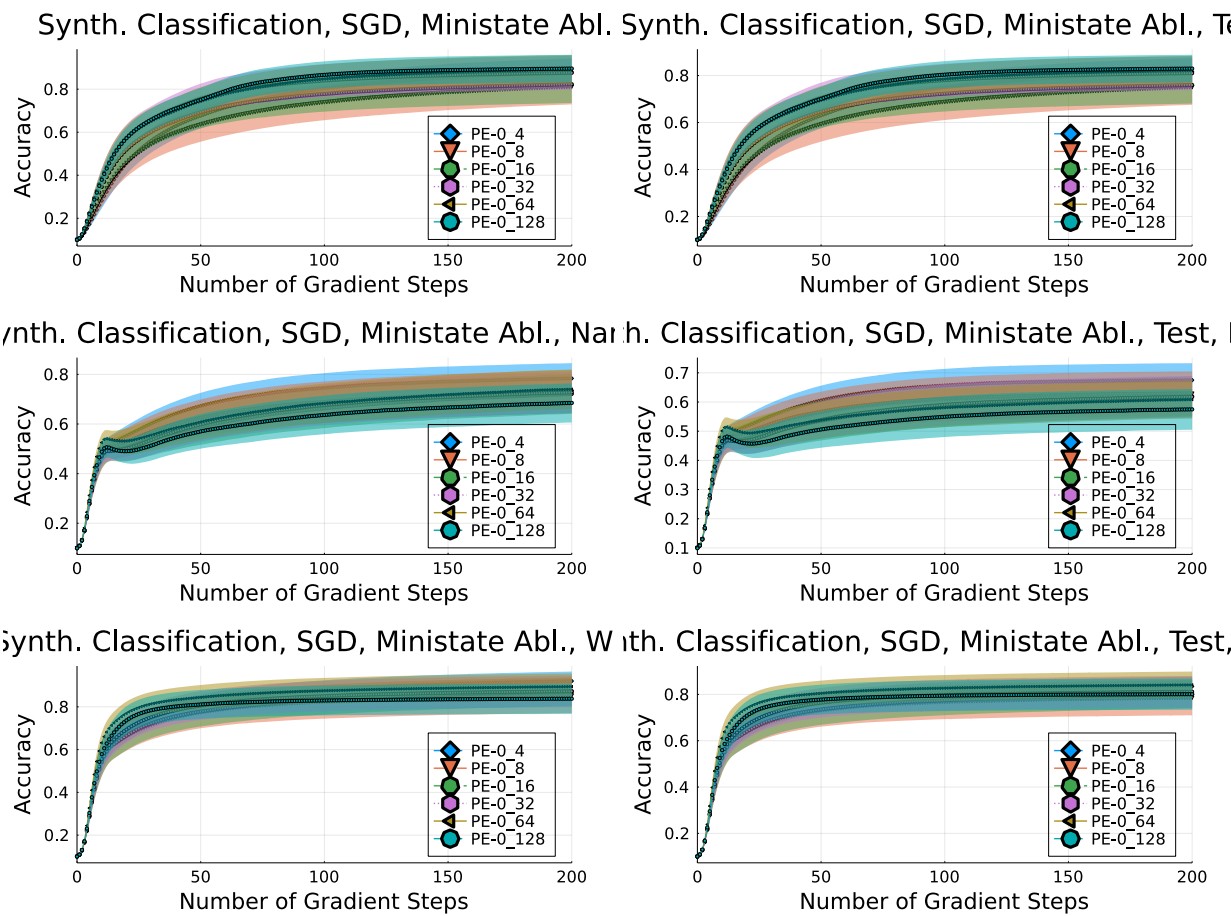

Figure 23: Synthetic Classification, Adam, Ministate size Ablation. Student training trajectories with a trained teacher. Left is training accuracy, right is testing accuracy. From top to bottom: same architecture as training, narrower architecture but deeper, wide architecture but shallower.

## L.4 Training Curves from Reward Ablation

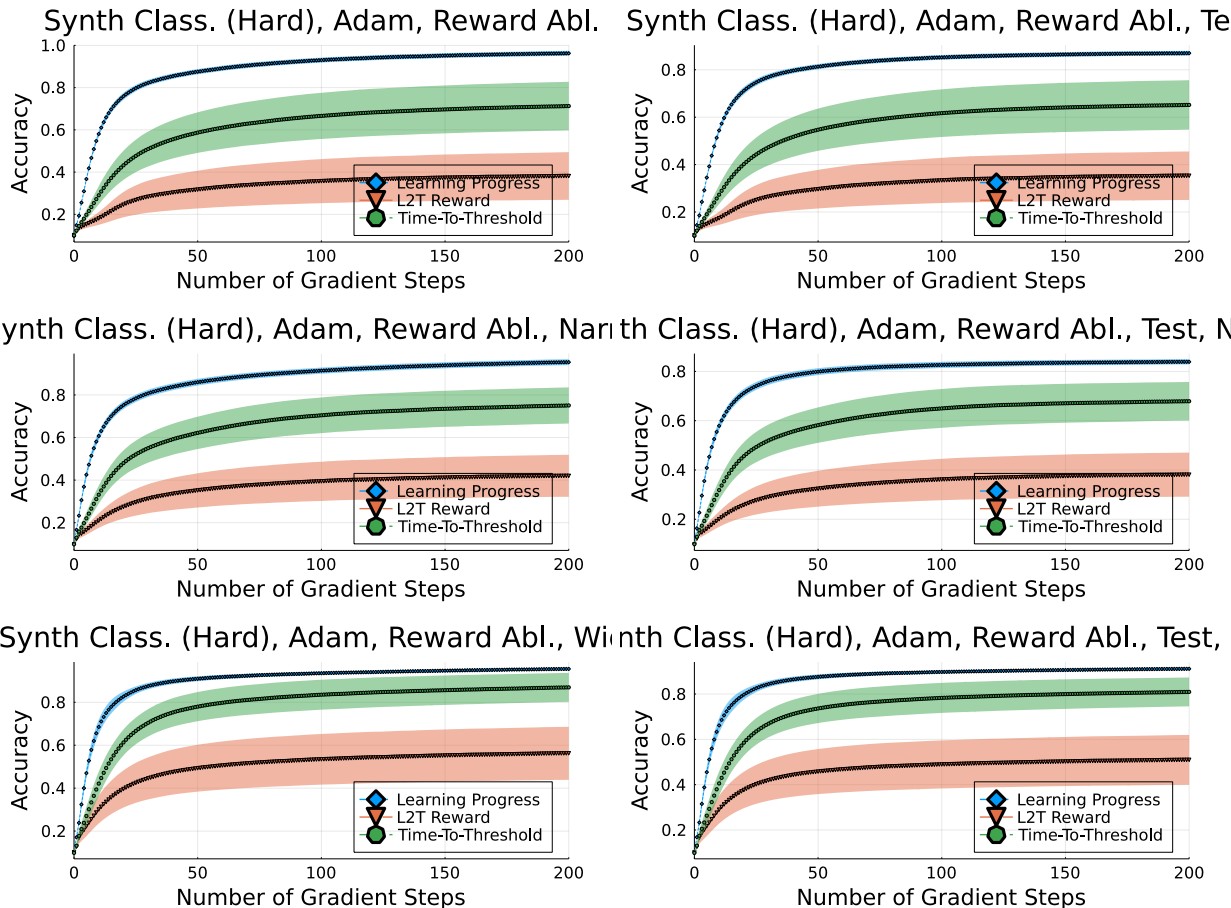

Figure 24: Synthetic Classification, Adam, Reward Ablation. Student training trajectories with a trained teacher. Left is training accuracy, right is testing accuracy. From top to bottom: same architecture as training, narrower architecture but deeper, wide architecture but shallower.

## L.5 Training Curves from Pooling Ablation

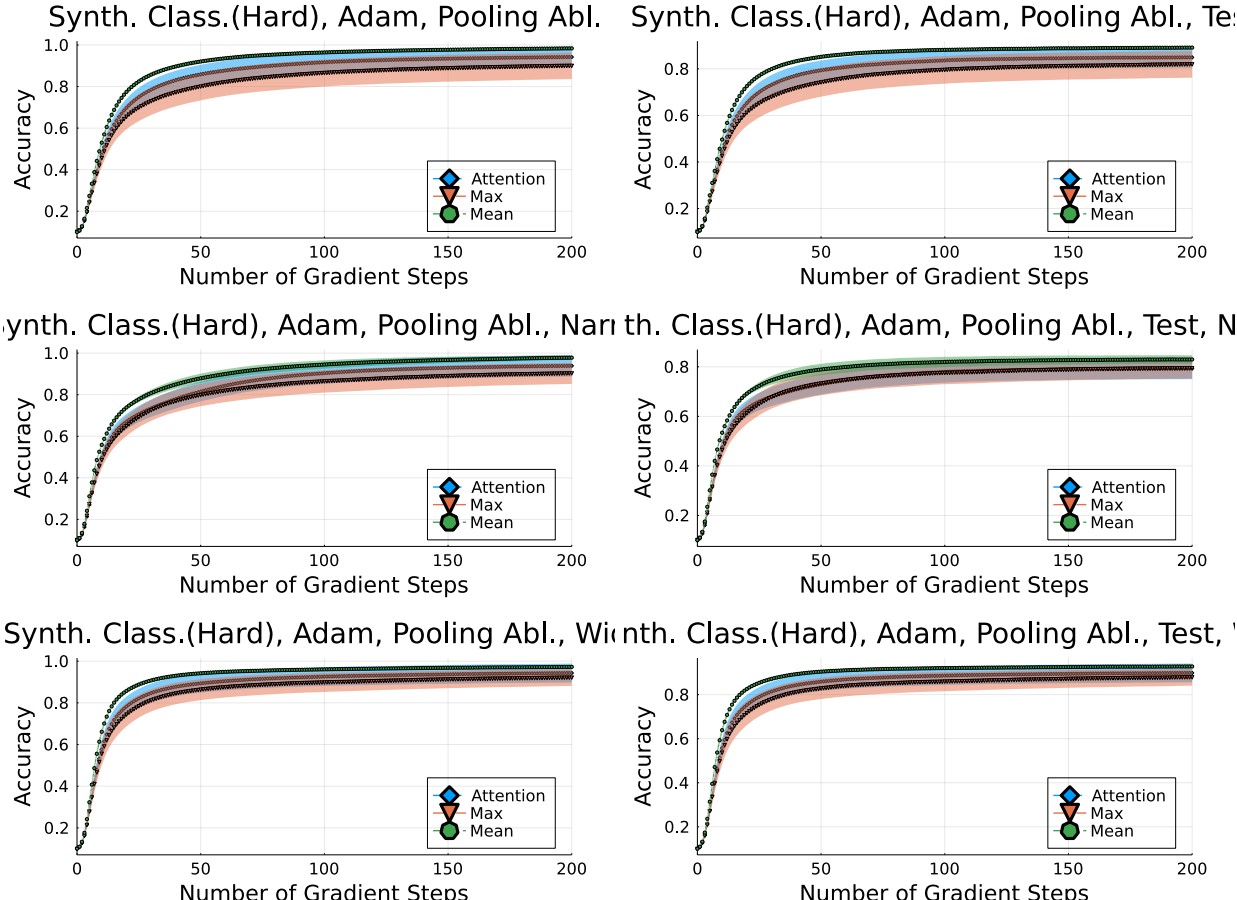

Figure 25: Synthetic Classification, Adam, Pooling Ablation. Student training trajectories with a trained teacher. Left is training accuracy, right is testing accuracy. From top to bottom: same architecture as training, narrower architecture but deeper, wide architecture but shallower.

## L.6 Training Curves from Synthetic NN Transfer Gym

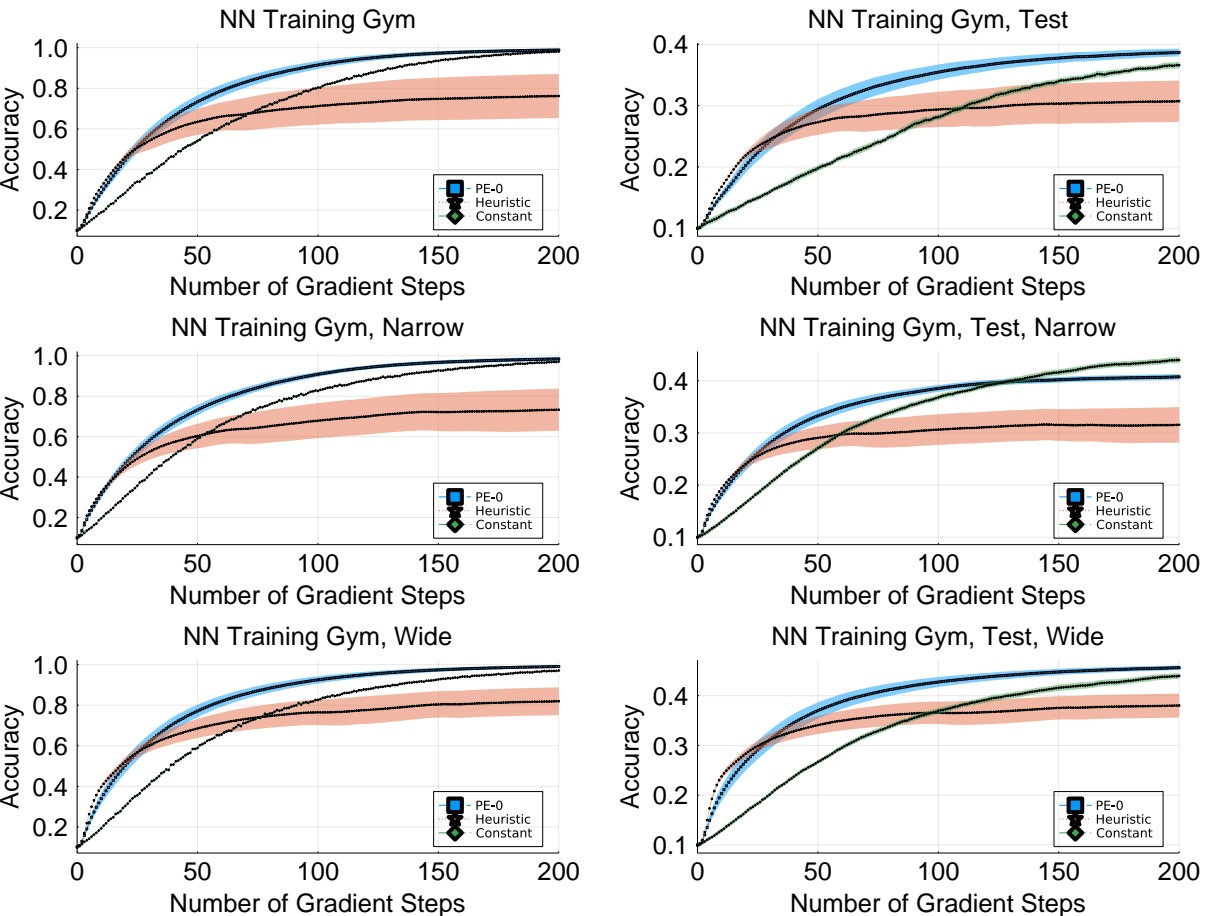

Figure 26: Transfer Gym Experiment using Adam as the base optimizer. Student training trajectories with a trained teacher. Left is training accuracy, right is testing accuracy. From top to bottom: same architecture as training, narrower architecture but deeper, wide architecture but shallower.

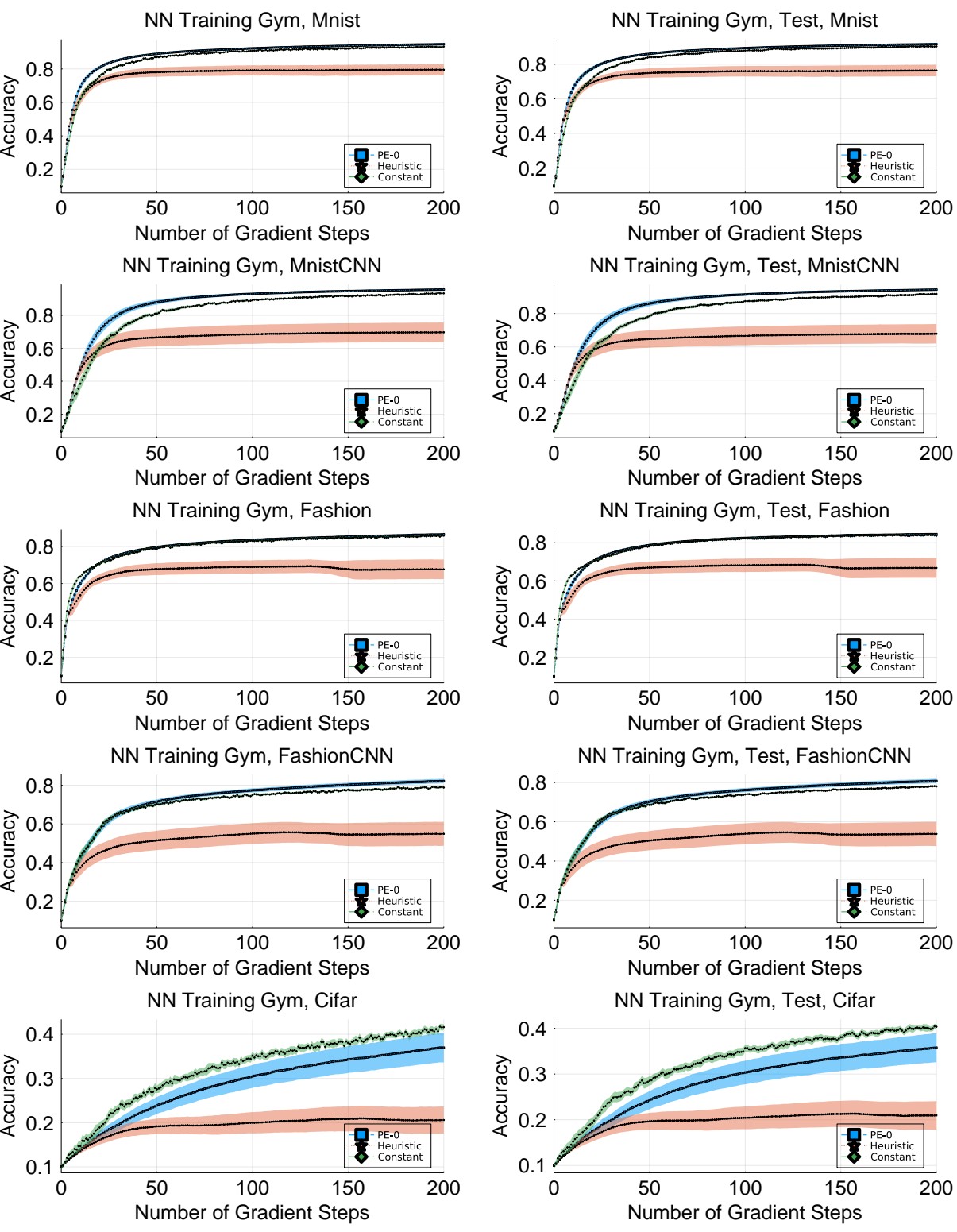

Figure 27: Transfer Gym Experiment using Adam as the base optimizer. Student training trajectories with a trained teacher. Left is training accuracy, right is testing accuracy. From top to bottom: Transfer to MNIST, Transfer to MNIST and CNN, transfer to Fashion MNIST, transfer to Fashion MNIST and CNN, transfer to CIFAR and CNN.

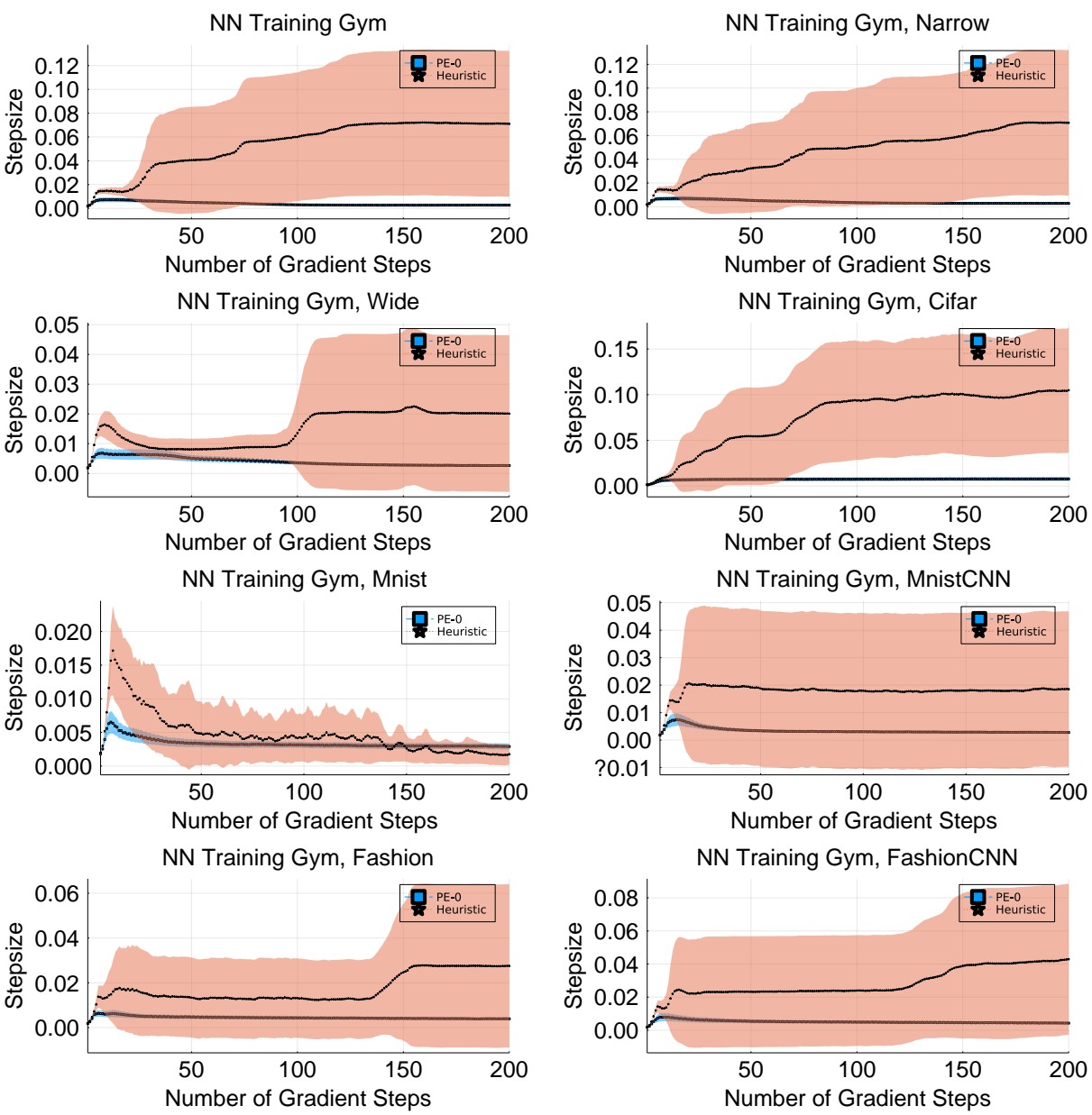

Figure 28: Transfer Gym Experiment using Adam as the base optimizer. Stepsizes selected by a trained teacher.

## L.7 Additional Experiment: Accuracy-Based Reward

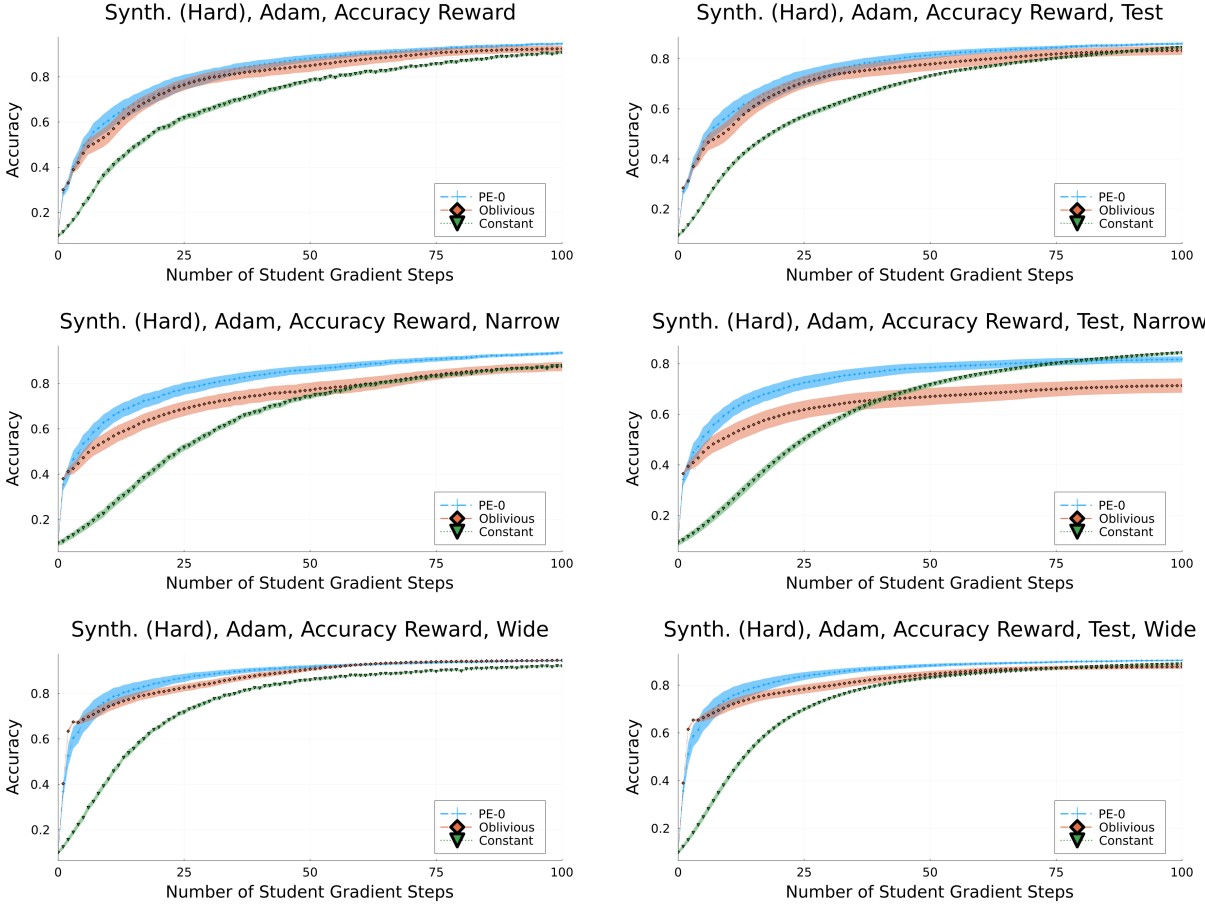

Figure 29: Hard Synthetic Classification expeirment using Adam as the base optimizer. Instead of time-to-threshold, the reward is just the accuracy at that time step. There is no termination. Student training trajectories with a trained teacher. Left is training accuracy, right is testing accuracy. From top to bottom: same architecture as training, narrower architecture but deeper, wide architecture but shallower.

