# OpenReview forum: "Reinforcement Teaching "
_TMLR — Accepted by TMLR_

### Review · Reviewer_HwBd · 2022-11-24

**Summary Of Contributions:**

The paper proposes a methodology for meta-learning by framing it in the MDP framework. The proposed method, called Reinforcement Teaching, consists of two components: student and teacher. The student is a base-level parametric optimization algorithm that learns to solve a task using some performance measure. The teacher is the meta-level optimization algorithm that takes the parameters of the base-level student algorithm as input with the aim of updating them so that the base-level algorithm performs better at the task. The paper proposes a method to represent the parameters of the student algorithm via an embedding scheme (Parametric-behavior Embedder) that when combined with classical RL reward-shaping techniques lead to better learning efficiency for the downstream student task. The approach is demonstrated on two tasks: curriculum generation in RL and synthetic classification via SGD.


**Audience:**

Yes

**Broader Impact Concerns:**

The authors have not acknowledged some of the limitations of their approach such as potential side effects that can arise due to a lack of guarantees for robustness.

**Claims And Evidence:**

No

**Requested Changes:**

- (Critical) Please check the comments regarding clarity and unsupported claims in the weakness section.

- (Critical) What if the teacher attempting to improve the student’s parameters drives the student learning algorithm to an unrecoverable state, for instance, a state where the student cannot ever achieve the desired threshold for the performance measure? This can happen in practice but we don’t see it in the experiments. Are there any reasons for this?

- (Minor) Why is hyper-parameter adaptation a sequential meta-learning task? One might argue that it is better to approach this task in a distributed/parallel computing manner.


**Strengths And Weaknesses:**

## Strengths:
The paper focuses on an important problem and the proposed methodology is straightforward. The paper provides experiments on both supervised learning and RL tasks.

## Weaknesses:

The main weakness of the paper is the lack of clarity and some unjustified statements. Here is the list of some of the points that I think require more accompanying details:

- The paper says that their approach is the first attempt at this problem but then they compare their method with baselines that also tackle similar problems. There is not enough clarity on the goals and scope of the work. For instance, the work is very closely related to gradient-based meta-learning approaches but why they cannot be applied to this problem is not explained in detail. It is mentioned in brief in the Introduction however a more thorough explanation is required. This is reflected later in the empirical studies too: the curriculum generation task can be treated in a hierarchical method as in goal-conditioned RL sense that does not require access to differentiable environment dynamics. Why the literature and methods from that literature are not considered is not given enough justification.

- For Section 4.4, it is not clear what the Action space looks like for the formulated MDP. The reader is referred to Appendix C that only contains a block diagram without any accompanying details. I think it will make sense to include some detailed examples that specify how the action spaces will look for the meta-MDP. This can be done by first formally defining the task, then describing the MDP that shows the correspondence with the original task.

- Section 5.1: The evaluation scheme is to evaluate the teacher’s policy on a new student, i.e., evaluate the curriculum sequencing for a new target position. But there is no mention of the target task distribution that is used for learning vs evaluation. Furthermore, the figures report the learning performance of the student which is apparently different from the evaluation scheme (Fig. 4) . The goal of the paper is to show the benefits of their method for better generalization and data efficiency but not many details are provided for these aspects.

- Is the teacher state Markovian?: The major contribution of the paper is the parametric-behaviour embedder that allows for a Markovian state representation. This is also the main distinguishing feature from the previous work in Learning to Teach Using RL (Section 3). However, I could not find any evidence of the claim that their method results in a Markovian state which allows them to treat the problem as an MDP. While the student’s parameters are Markovian, when a bunch of such markovian states are combined within the embedding module it is not guaranteed that the teacher’s state will be Markovian. The main argument made by the authors is that the local information provided by student behaviour (for a large enough minibatch) is able to maintain the Markovian property. Maybe I’m missing something but I don’t see why the claim is true. Note that the teacher's state might not be Markovian and that's fine too from an algorithmic perspective. My concern is more regarding if the claim is supported or not.


- More details on experimentation methodology: The empirical studies have different methodologies where the baselines considered for both of them are different, as well as the plots and evaluation metrics are also different. There is no justification on why some baselines cannot be used for other methods, e.g., why cannot PVN be used for the curricula generation task? Additionally, I believe it will be useful to first formally define the meta-MDP of the considered domains and the learning problems before describing the methodology.

---

> ### Author Response · Authors · 2023-02-07
> **Response to reviewer HwBd (1/4)**
>
> Dear reviewer HwBd,
>
> Thank you for your thorough review and for providing helpful feedback. We now will address the weaknesses and requested changes below. We have also made several changes in the manuscript which are highlighted in blue.
>
> Weakness 1a: "The paper says that their approach is the first attempt at this problem but then they compare their method with baselines that also tackle similar problems. There is not enough clarity on the goals and scope of the work."
>
> Response: The objective of our work is to develop a framework that can improve machine learning algorithms across diverse settings. For example, it’s possible to improve the efficiency of RL algorithms by learning curricula. It’s also possible to improve ML algorithms by performing hyperparameter adaption. However, from surveying the literature, current works typically focus on improving ML algorithms in a specific manner (i.e., by only developing curricula or by only performing hyperparameter adaption). The goal of these works and their respective solution methods are not general approaches that can improve ML algorithms in more than one manner. For example, the Narvekar et al. (2017) baseline (e.g., student parameters with time-to-threshold reward) was initially designed to learn curricula. The intention and the design of our Reinforcement Teaching framework is that of a general formulation that can be applied to different problem settings. Therefore, in our empirical results, we showed that Reinforcement Teaching, unlike other RL-teaching baselines, can be effective in more than one problem setting (e.g.,  curriculum learning and step-size adaption).
>
> Weakness 1b: "For instance, the work is very closely related to gradient-based meta-learning approaches but why they cannot be applied to this problem is not explained in detail."
>
> Response: We will first briefly describe the relationship between gradient-based meta-learning and Reinforcement Teaching, before outlining why gradient-based meta-learning cannot be applied to our experiments. In the language of Reinforcement Teaching, gradient-based meta-learning algorithms are analogous to model-based trajectory optimization, where the state-transition model is the gradient update. Model-based trajectory optimization, however, is only applicable if 1) we have the state-transition model, 2) that state-transition model is fully-differentiable and 3) planning an entire trajectory using the fully-differentiable model is computationally feasible. In our reinforcement learning experiments, the state-transition model is not just a gradient-update but a sequence of interactions between the student’s policy and its environment. While the gradient-update itself is differentiable, the interaction between the student’s policy and the environment is not differentiable. Hence, the state-transition model is not differentiable, making gradient-based meta-learning inapplicable.  For our supervised learning experiments, gradient-based meta-learning is applicable in principle because the state-transition model is just a gradient-update. As discussed in Section 2, however, gradient-based optimization requires extensive trajectory-based optimization that is specific to each state and student or architecture. Reinforcement teaching allows us to learn a policy that can be queried quickly at any step and for any student or architecture, but gradient-based meta-learning requires expensive re-computation at each time-step. Hence gradient-based meta-learning is computationally infeasible, and less generalizable to changes in the meta-learning problem, compared to Reinforcement Teaching. In the revised submission, we have added clarification to Section 2) and have added a new section, Appendix G: “Connecting Reinforcement Teaching to Gradient-Based Meta-Learinng”.
>
> Weakness 2: "For Section 4.4, it is not clear what the Action space looks like for the formulated MDP. The reader is referred to Appendix C that only contains a block diagram without any accompanying details. I think it will make sense to include some detailed examples that specify how the action spaces will look for the meta-MDP. This can be done by first formally defining the task, then describing the MDP that shows the correspondence with the original task."
>
> Response: In Sections 5.1 and 5.2, we have added a new “Teaching MDP” paragraph that outlines in more detail the respective meta-MDP for the curriculum learning and step-size adaption problem settings. These paragraphs discuss how the actions, states, and rewards are initialized. We also included a new table (See Table 1) that provides an overview of the teaching MDP initializations for both problem settings.

---

> > ### Author Response · Authors · 2023-02-07
> > **Response to reviewer HwBd (2/4)**
> >
> > Weakness 3: "Section 5.1: The evaluation scheme is to evaluate the teacher’s policy on a new student, i.e., evaluate the curriculum sequencing for a new target position. But there is no mention of the target task distribution that is used for learning vs evaluation. Furthermore, the figures report the learning performance of the student which is apparently different from the evaluation scheme (Fig. 4) . The goal of the paper is to show the benefits of their method for better generalization and data efficiency but not many details are provided for these aspects."
> >
> > Response: For the curriculum learning experiments, the teacher’s objective is to learn the best sequencing of sub-tasks such that the student can solve its respective target task as quickly as possible. For example, in the Maze and Four Room environments, the sub-tasks correspond to different environment start states. In both cases (i.e., Maze and Four Rooms), the student’s target task is to travel from the blue state to the green state (See Appendix H.1, Figures 9 and 10). The blue state is the target start state and the green state is the target terminal state. During the teacher’s training, the teacher has the choice of 11 start states (including the target start state) for the Maze environment and 10 start states  (including the target start state)  for the Four Rooms environment. For each teacher episode, we start with a newly initialized student agent. To begin, the teacher chooses a start state (i.e., an action) and the student is then placed at that location. The student’s goal is to learn how to travel from that start state specified by the teacher to the target terminal state. The student will train for a given number of episodes with this specific subtask.  This is considered one time step from the teacher’s perspective. Once the student’s training is complete, we retrieve the teacher’s respective state and reward and update the teacher’s policy. Afterwards, the teacher selects another start state and this process continues. This process continues for a maximum number of teacher-time steps. After this maximum is reached, a new teacher episode begins, and we obtain a newly initialized student agent once more.
> >
> > A similar process occurs during the teacher’s evaluation. The only two differences are: (1) in evaluation we do not update the teacher’s policy and (2): there is only one teacher episode in which the teacher is assigned a newly initialized student. The teacher does not have multiple episodes to produce a curriculum. During evaluation, the student’s target task remains the same as in training: travel from the blue target start state to green target terminal state. There is not a difference in target task distribution that is used for training vs. evaluation.
> >
> > For the curriculum learning section, our goal is to determine whether the RL teacher, using our parametric embedder state representation and our LP reward function, can learn a curriculum that results in faster student learning on the target task. In Figure 4-right, for every student episode on the X-axis, the teacher provides a subtask (i.e., one of the starting states). The student then trains with that subtask. Afterward, we report the student’s performance on the target task on the Y-Axis. Therefore, Figure 4-right shows that by using the curriculum learned by our RL teacher, the student learns the target task more efficiently.
> >
> > To make this more clear in our submission, we have added two separate paragraphs in Section 5.1 and Section 5.2, titled “Teaching Training” and “Teaching Evaluation”. These paragraphs further describe the training and evaluation protocol of the teacher.  We have also moved the pseudocode of the teacher’s training process from Appendix B to the main text. Please see Algorithm 1.
> >
> > Weakness 4: "Is the teacher state Markovian?... Note that the teacher's state might not be Markovian and that's fine too from an algorithmic perspective. My concern is more regarding if the claim is supported or not"
> >
> > Response: It is correct that the parameter-behavior embedder does not necessarily satisfy the Markov property when training with SGD. We have relaxed the Markov argument in our submission (See updated Section 4.2.1), and instead, provide an empirical argument showing that the Parametric-Behavior embedder is still able to provide a state-representation that improves over the parameter-based state representation (which is Markov). We hypothesize that, although the parametric-behavior embedder is not necessarily Markov across the entire student function space, it provides a useful state abstraction on the subspace of functions encountered during the student’s gradient-based training.

---

> > > ### Author Response · Authors · 2023-02-07
> > > **Response to reviewer HwBd (3/4)**
> > >
> > > Weakness 5: "More details on experimentation methodology: The empirical studies have different methodologies where the baselines considered for both of them are different, as well as the plots and evaluation metrics are also different. There is no justification on why some baselines cannot be used for other methods, e.g., why cannot PVN be used for the curricula generation task? Additionally, I believe it will be useful to first formally define the meta-MDP of the considered domains and the learning problems before describing the methodology."
> > >
> > > Response: The supervised learning and reinforcement learning experiments both aim to demonstrate the generality of reinforcement teaching. However, reinforcement learning and supervised learning are quite different problem settings. While the reinforcement teaching framework aims to be general, the experimental methodology (baselines, etc.) is tailored to each setting (supervised vs reinforcement student).
> > >
> > > To better unify the experiments, we have added a “Teacher Evaluation” paragraph in Sections 5.1 and Section 5.2 to the submission to better describe the evaluation in terms of (1) the evaluation metrics (e.g., student and teacher learning curves) and (2) the baselines used in both curriculum learning and step-size adaption settings. Moreover, to make the evaluation results more cohesive across both problem settings, we have (1) added the teacher learning curves for the Curriculum Learning setting (See Section 5.1 Figure 4-left) and (2) moved the student learning curves for the step-size adaptation setting from Appendix K to the main text (See Section 5.2 Figures 5-right and 6-right). By doing so, the evaluation plots should now be similar across both settings. In addition, we note that the visualization for the state and reward ablation results are different across the curriculum learning (table format showing area under student learning curves) and step-size adaptation setting (plot showing student learning curves). The primary reason for this was for the sake of limiting space. Tables can condense information and take up less space per page compared to a figure.
> > >
> > > As for differences between the baselines used for the two problem settings, we first note that TSCL was only used for the curriculum learning setting because we wanted to compare against another curriculum learning specific baseline that took an alternate bandit approach. For the curriculum learning state ablation, we will add to our submission a new result for the PVN state representation. In the supervised learning experiments, the baselines are chosen to be relevant towards each ablation. For the state-representation ablation, we include all discussed state-representations using the time-to-threshold reward. Because PVN and Parameters perform so poorly and have a high memory and computation burden, we do not include them in further experiments. In place of these state-representations, we also include the performance of Adam with a tuned constant learning rate because tuning Adam is ubiquitous and effective in practice.
> > >
> > > Requested changes 1: "Please check the comments regarding clarity and unsupported claims in the weakness section."
> > >
> > > Response: Our response to the unsupported claims and lack of clarity are reflected in our previous responses to weaknesses 1-5. In addition, as discussed in our earlier responses, we have made significant additions to our submission to improve the overall clarity and accuracy of claims.

---

> > > > ### Author Response · Authors · 2023-02-07
> > > > **Response to reviewer HwBd (4/4)**
> > > >
> > > > Requested changes 2: "What if the teacher attempting to improve the student’s parameters drives the student learning algorithm to an unrecoverable state, for instance, a state where the student cannot ever achieve the desired threshold for the performance measure? This can happen in practice but we don’t see it in the experiments. Are there any reasons for this?"
> > > >
> > > > Response: For every teacher episode, the teacher encounters a newly initialized student. For each teacher time-step, the teacher will first take an action (e.g., select a subtask, select a step-size). Then this action will update a part of the student’s learning process. For example, with the step-size adaptation setting, the teacher’s action then updates the step size of the student’s optimizer. The student will then take a gradient step with that respective step-size. Afterward, the teacher observes its next state and reward. This interaction continues until the student reaches its performance threshold or until the maximum number of teacher time steps has been reached, whichever comes first. During a teacher episode, the teacher can take a bad action that leads the student to an unrecoverable state.
> > > >
> > > > Although this occurs, the teacher will continue interacting with the student (e.g., by selecting step-size or subtask) until the maximum number of teacher time steps has been reached. Once this happens, the teacher episode will end, and a new teacher episode will begin with a newly initialized student (i.e., the student is reset). Since we reset the student at the start of every teacher episode, the teacher gets a new attempt at improving the student, and since it likely received negative reward for whichever poor actions it previously took (for which is now stored in the teacher’s replay buffer), the teacher will avoid making that mistake again.
> > > >
> > > > Requested changes 3: "Why is hyper-parameter adaptation a sequential meta-learning task? One might argue that it is better to approach this task in a distributed/parallel computing manner."
> > > >
> > > > Response: Hyper-parameter adaptation is different from the problem of hyper-parameter tuning, for which distributed computing is used. In adaptation, the teacher/meta-learner must increment or decrement a hyperparameter in response to the student’s current learning ability. In tuning, we are searching over the best fixed hyperparameter irrespective of a particular learning algorithm’s roll-out. Section 2 provides some discussion on this, and we have added further context for hyper-parameter adaptation v.s. Distributed computing.
> > > >
> > > > Broader Impact Concerns: "The authors have not acknowledged some of the limitations of their approach such as potential side effects that can arise due to a lack of guarantees for robustness."
> > > >
> > > > Response: Our framework is general in its application, but this generality makes it difficult to assess formal guarantees of robustness. Because we use reinforcement learning, we could potentially leverage previous results on robustness in idealized settings. Unfortunately, characterizing the teaching MDP formally is difficult because the state space (parameters) does not clearly adhere to current classifications (e.g. tabular, linear, low-rank). Rigorous theoretical analysis, however, is an interesting direction for further research.

---

> ### Author Response · Authors · 2023-04-21
> **Follow-up on Reviewer HwBd**
>
> We are following up on this review with a very brief summary of our rebuttal with regards to your three requested changes:
>
> - "Please check the comments regarding clarity and unsupported claims in the weakness section."
> Response: We have added a few clarifying details to our contributions in light of reviewer vBun: Reinforcement Teaching enables the use of RL for meta-learning with non-linear students across different problem settings. We have also added details in Section 2 and the Appendix on why gradient-based meta-learning cannot always be used. We would also like to point the reviewer to the following excerpt from another rebuttal comment on clarifying the assumptions we are making in setting up Reinforcement Teaching:
>
> ```
> In setting up the teaching MDP, we are only assuming there is a given meta-learning problem to solve. The meta-learning problem defines how we can change the student (the action) and the desired goal for the student (the reward). The choice of the reward function is not given explicitly, but it is defined implicitly in the meta-learning problem's goal. In curriculum learning for RL, the meta-learning problem is getting an agent to solve a hard task quickly by sequencing tasks. The time-to-threshold reward is a faithful representation of this goal.
> ```
>
> - "What if the teacher attempting to improve the student’s parameters drives the student learning algorithm to an unrecoverable state,"
> Response: Similar to a reinforcement learning algorithm interacting in an episodic environment, this can and does happen in Reinforcmeent Teaching but it does not prevent the teacher from learning to recover from these failures. The teacher can drive the student learning algorithm to an unrecoverable state for an episode. But, on the next episode the student algorithm is reset and the teacher is allowed to try again. The experience of driving the student to an unrecoverable state is stored in a replay buffer so the teacher can learn to not make this error again.
>
> - "Why is hyper-parameter adaptation a sequential meta-learning task? One might argue that it is better to approach this task in a distributed/parallel computing manner."
> Response: We have added some details to section 2 to clarify that hyperparameter adaptation is a separate problem from hyperparameter tuning. In the former, the goal is to find a hyperparameter sequence. The goal in the latter is to find the best fixed hyperparameter, which can benefit from distributed/parallel computing.

---

> > ### Comment · Reviewer_HwBd · 2023-04-28
> > **Thank you for the response!**
> >
> > My concerns related to the connection gradient-based meta-learning approaches and action descriptions have been addressed. I would recommend the authors provide a link to Appendix H in the main text too. The new sections on Teacher training and evaluation provide much-needed clarity on the experimentation methodology. Overall, I think the paper has definitely improved, both in terms of clarity and accuracy of the claims.

---

### Review · Reviewer_26c6 · 2022-12-02

**Summary Of Contributions:**


This paper makes the following contributions:

- It introduces and formalizes the Reinforcement Teaching problem.
- It proposes a way to model this as an MDP and embed the learners neural network to use as the state of the MDP.
- Tests Reinforcement Teaching on a curriculum learning and step size learning task and shows good performance compared to baselines and ablations of the proposed approach.




**Audience:**

Yes

**Claims And Evidence:**

Yes

**Requested Changes:**

What would happen if you used m(theta) - m* as the reward function? It seems that this might be better shaped since it doesn't rely on the potentially noisier change in performance between two time steps? I would like to see some results comparing this reward to the others in the paper.

I like how Reinforcement Teaching is a nice unifying problem setting.

In Figure 3, why was the top left starting state and other starting states never chosen?

Why use DQN for the teacher's policy? What would be the effect of using a differnt policy?

I like that the authors consider embedding values and embedding actions. It’s unclear how the dataset is chosen for collecting the values or actions, though. This should be made more clear

What is the reason for using Double DQN for the stepsize adaptation vs DQN for the RL? These choices seem arbitrary and some discussion and experimental analysis would be useful to know when one is better than the other.

The authors point out that fixed probing inputs are insufficient for representing neural networks. This makes me wonder how the inputs for the PE embeddings are chosen. Are they chosen randomly? Could they be learned or adapted or selected actively according to some metric?

I’m confused about why PE-O works the best in Figure 5. It seems like a representation just based on outputs would not tell much without knowing what the inputs were. Some intuition or experiment explaining this would be beneficial.

On a related note, are the outputs the maximum probability class labels or a softmax distribution.


Missing references:

It would be good to mention the following work on machine teaching for sequential decision making tasks when talking about machine teaching:

- Brown, Daniel S., and Scott Niekum. "Machine teaching for inverse reinforcement learning: Algorithms and applications." Proceedings of the AAAI Conference on Artificial Intelligence. Vol. 33. No. 01. 2019.

The idea of reinforcement teaching also seems very related to reward design methods and it would be good to discuss how these works relate. It seems that reward/cost design could be viewed as a special case of reinforcement teaching where there the teacher optimizes the reward given to the learner in order to improve performance:

- Singh, Satinder, Richard L. Lewis, and Andrew G. Barto. "Where do rewards come from." Proceedings of the annual conference of the cognitive science society. Cognitive Science Society, 2009.

- Sorg, Jonathan, Richard L. Lewis, and Satinder Singh. "Reward design via online gradient ascent." Advances in Neural Information Processing Systems 23 (2010).

- Jain, A., Chan, L., Brown, D. S., & Dragan, A. D. (2021, May). Optimal Cost Design for Model Predictive Control. In Learning for Dynamics and Control (pp. 1205-1217). PMLR.





**Strengths And Weaknesses:**

Strengths:
- The results on transferring the stepsize adaptation policy are nice and show the effectiveness of the approach.
- I like that the authors consider embedding values and embedding actions.
- I like how Reinforcement Teaching is a nice unifying problem setting.
- The paper nicely addressses shaping rewards in 4.3.1

Weaknesses
- There were several places that did not have enough details (see below)
- The efficiency of the method has be slightly concerned since RL often struggles with efficiency and how we have meta RL controlling RL (at least for one of the domains).
- Some design choices are not supported. Especially the choice of RL algorithms for Reinforcement Teaching

---

> ### Author Response · Authors · 2023-02-07
> **Author's response to Reviewer 26c6 (1/3)**
>
> Dear reviewer 26c6,
>
> Thank you for providing helpful suggestions. We now will address the weaknesses and requested changes below. We have also made several changes in the manuscript which are highlighted in blue.
>
> Weakness 2: "The efficiency of the method has be slightly concerned since RL often struggles with efficiency and how we have meta RL controlling RL (at least for one of the domains)."
>
> Response: Learning efficiency is indeed a common issue with RL, particularly when the reward is sparse. In Appendix F we outline the difficulties with RL-teaching methods in terms of the sample complexity. Because of this, however, we added experimental results showing the teacher’s own learning efficiency. See Section 5.1, Figure 4-left, and Section 5.2 Figure 5-left, and Figure 6-left. We found that with our Reinforcement Teaching method (using PE state + LP reward), the teacher is able to learn an effective policy more quickly compared to other RL-teaching methods that use different state representations and reward functions.
>
> Requested Changes 1: "What would happen if you used m(theta) - m* as the reward function? It seems that this might be better shaped since it doesn't rely on the potentially noisier change in performance between two time steps? I would like to see some results comparing this reward to the others in the paper."
>
> Response: We thank the reviewer for this suggestion. We will add results with this new reward function to our reward ablation experiments.
>
> Requested Changes 3: "In Figure 3, why was the top left starting state and other starting states never chosen?"
>
> Response: For the Maze curriculum learning experiments, the teacher’s objective is to learn a sequence of starting states for the student such that the student learns how to solve the target task as quickly as possible. The student’s target task is to travel from the blue start state to the green terminal state. See Appendix H.1, Figure 9 for an image of the Maze target task. The starting states at coordinates (1,1), (5,1), (9,1), and (7,13) are likely never chosen by the teacher because they are not in the most efficient path from the blue to green state. Therefore, if the teacher would have included these starting states in its curriculum, they would not have provided additional help for the student on its target task. From the teacher’s perspective, the teacher would have then received an additional negative reward (-1 + Learning Progress) for each un-necessary starting state it included.
>
> Requested Changes 4: "Why use DQN for the teacher's policy? What would be the effect of using a differnt policy?"
>
> Response: This is ultimately a design choice, and we chose DQN and DDQN for the teacher’s policy for two primary reasons. First, we wanted our Reinforcement Teaching framework to have lower sample complexity and DQN is a long-established deep off-policy RL algorithm. In Appendix G, we discussed that interacting with the teacher’s MDP and evaluating the teacher can be expensive. This is because each teacher episode corresponds to an entire training trajectory of the student. Off-policy learning is generally more sample efficient than on-policy methods because of its ability to reuse past experiences that are stored in the replay buffer, thus the family of DQN algorithms is one natural choice.
>
> Second, our goal is to evaluate the efficacy of our Reinforcement Teaching framework in solving multiple meta-learning problems. Although there have been advancements in off-policy learning algorithms [1-3] and these improvements will likely improve the overall performance of our framework, we wanted to study Reinforcement Teaching in the simplest (deep RL) setting possible. In the original submission, we included an explanation of using DQN/DDQN in Appendix G. However, we have added a new paragraph, “Teacher Training” to Section 5.1 and Section 5.2 to further elaborate on these design choices.

---

> > ### Comment · Reviewer_26c6 · 2023-04-26
> > **Unaddressed concerns**
> >
> > Thank you for the response. I appreciate the clarification and further experiments. However, many of my questions and requested revisions have not been addressed. I would like the authors to address the following (copied from my original review):
> >
> >
> > I like that the authors consider embedding values and embedding actions. It’s unclear how the dataset is chosen for collecting the values or actions, though. This should be made more clear.
> >
> > What is the reason for using Double DQN for the stepsize adaptation vs DQN for the RL? These choices seem arbitrary and some discussion and experimental analysis would be useful to know when one is better than the other.
> >
> > The authors point out that fixed probing inputs are insufficient for representing neural networks. This makes me wonder how the inputs for the PE embeddings are chosen. Are they chosen randomly? Could they be learned or adapted or selected actively according to some metric?
> >
> > I’m confused about why PE-O works the best in Figure 5. It seems like a representation just based on outputs would not tell much without knowing what the inputs were. Some intuition or experiment explaining this would be beneficial.
> >
> > On a related note, are the outputs the maximum probability class labels or a softmax distribution?
> >
> >
> > Missing references:
> >
> > It would be good to mention the following work on machine teaching for sequential decision making tasks when talking about machine teaching:
> >
> > - Brown, Daniel S., and Scott Niekum. "Machine teaching for inverse reinforcement learning: Algorithms and applications." Proceedings of the AAAI Conference on Artificial Intelligence. Vol. 33. No. 01. 2019.
> >
> > The idea of reinforcement teaching also seems very related to reward design methods and it would be good to discuss how these works relate. It seems that reward/cost design could be viewed as a special case of reinforcement teaching where there the teacher optimizes the reward given to the learner in order to improve performance:
> >
> > - Singh, Satinder, Richard L. Lewis, and Andrew G. Barto. "Where do rewards come from." Proceedings of the annual conference of the cognitive science society. Cognitive Science Society, 2009.
> >
> > - Sorg, Jonathan, Richard L. Lewis, and Satinder Singh. "Reward design via online gradient ascent." Advances in Neural Information Processing Systems 23 (2010).
> >
> > - Jain, A., Chan, L., Brown, D. S., & Dragan, A. D. (2021, May). Optimal Cost Design for Model Predictive Control. In Learning for Dynamics and Control (pp. 1205-1217). PMLR.

---

> > > ### Author Response · Authors · 2023-04-26
> > > **Follow up to unaddressd concerns**
> > >
> > > Thanks for your response!
> > >
> > > We believe we have addressed your concerns in the official comments "Author's response to Reviewer 26c6 (2/3)" and "Author's response to Reviewer 26c6 (3/3)". Are you able to see those responses? If not, I can include them again.
> > >
> > > Please let me know if those responses addressed your concerns.

---

> > > > ### Comment · Reviewer_26c6 · 2023-04-28
> > > > **Cannot see additional responses**
> > > >
> > > > No. I do not see the other responses. Not sure why. Can you please repost them?

---

> > > > > ### Author Response · Authors · 2023-04-28
> > > > > **Reposting responses**
> > > > >
> > > > > We copied them below!
> > > > >
> > > > > Requested Changes 5: "I like that the authors consider embedding values and embedding actions. It’s unclear how the dataset is chosen for collecting the values or actions, though. This should be made more clear"
> > > > >
> > > > > Response: During the student’s training process, the student encounters a set of input/output pairs. In the case of curriculum learning, the input/output pairs would correspond to (student state, student action) pairs. These pairs are then stored in a buffer. If the input (e.g., student state) is already in the buffer, we keep the latest output (e.g., student action) that was taken. When it’s time to retrieve the teacher’s state, we randomly sample a minibatch of M (student state, student action) pairs from this buffer. For the embedding actions state representation, this is all the information required. For the embedded value state representation, we query the state/action value corresponding to each student state using the most up-to-date value network. In our original submission, this information was primarily in Section 4.2.1. However, we have updated Section 5.1, the "Teaching MDP for Curriculum Learning" paragraph to include a better description of how the PE-Actions and PE-Values representations are formed.
> > > > >
> > > > > Requested Changes 6: "What is the reason for using Double DQN for the stepsize adaptation vs DQN for the RL? These choices seem arbitrary and some discussion and experimental analysis would be useful to know when one is better than the other."
> > > > >
> > > > > Response: In our experimental design, we wanted to use off-policy RL algorithms for the teacher because they tend to have lower sample complexity compared to the on-policy counterparts. Moreover, we wanted to demonstrate the effectiveness of our Reinforcement Teaching approach using the simplest of off-policy algorithms, therefore we used DQN and DDQN. We see the difference between DQN and DDQN as an implementation detail which is clear in the code that we have included in our submission. Therefore, we do not expect a significant difference in the teacher’s performance between these two algorithms in either the curriculum learning or step-size adaptation setting.
> > > > >
> > > > > Requested changes 7: "The authors point out that fixed probing inputs are insufficient for representing neural networks. This makes me wonder how the inputs for the PE embeddings are chosen. Are they chosen randomly? Could they be learned or adapted or selected actively according to some metric?"
> > > > >
> > > > > Response: The inputs for the PE embeddings are randomly selected from a buffer of past student inputs. Specifically, randomization is done each time the state is constructed. For some fixed student parameterization, each realization of the state from the PE embedding is different due to randomization. We hypothesize that this randomization forces the teacher to learn a more robust and generalizable policy. It would be an interesting future direction to consider different input sampling methods, such as those that prioritize certain inputs in the sampling procedure.
> > > > >
> > > > > Requested changes 8 and 9: "I’m confused about why PE-O works the best in Figure 5. It seems like a representation just based on outputs would not tell much without knowing what the inputs were. Some intuition or experiment explaining this would be beneficial. On a related note, are the outputs the maximum probability class labels or a softmax distribution."
> > > > >
> > > > > Response: In summary, PE-0 provides a state representation based on the softmax class probability prediction for an i.i.d. sample of images. Its relative success as a state representation suggests that learning is predominantly influenced by the average behavior of a classifier on a minibatch of images and less on outliers. This is in contrast to even coarser grained measures, such as the heuristic state representation which is the average loss on a minibatch.
> > > > >
> > > > > Requested changes 10: "Missing references. It would be good to mention the following work on machine teaching for sequential decision making tasks when talking about machine teaching:
> > > > >
> > > > > Response: We thank the reviewer for providing this additional related work. We will include this in the Machine Teaching paragraph of Section 3.

---

> > > > > > ### Author Response · Authors · 2023-04-28
> > > > > > **Reposting remaining responses**
> > > > > >
> > > > > > Requested changes 11: "The idea of reinforcement teaching also seems very related to reward design methods and it would be good to discuss how these works relate. It seems that reward/cost design could be viewed as a special case of reinforcement teaching where there the teacher optimizes the reward given to the learner in order to improve performance:"
> > > > > >
> > > > > > Response: We thank the reviewer for providing these new references. It does seem that reward/cost design can be unified under the Reinforcement Teaching framework. For this problem setting, the teacher would take actions that adjust the student’s reward/cost function during the student’s learning process. This is also related to adaptive reward shaping, in which the teacher can learn a policy to adaptively shape the student’s reward function to improve their overall learning. Recent work has seen some success in that area [4].
> > > > > >
> > > > > > References:
> > > > > >
> > > > > > [1] Timothy P. Lillicrap, Jonathan J. Hunt, Alexander Pritzel, Nicolas Heess, Tom Erez, Yuval Tassa, David Silver, and Daan Wierstra. Continuous control with deep reinforcement learning. Fourth International Conference on Learning Representations, 2016. URL https://openreview.net/forum?id=tX_O8O-8Zl
> > > > > >
> > > > > > [2] Tuomas Haarnoja, Aurick Zhou, Pieter Abbeel, and Sergey Levine. Soft actor-critic: Off-policy maximum entropy deep reinforcement learning with a stochastic actor. 2018. URL https://arxiv.org/pdf/1801. 01290.pdf
> > > > > >
> > > > > > [3] Scott Fujimoto, Herke van Hoof, and David Meger. Addressing function approximation error in actor-critic methods. Thirty fifth International Conference on Machine Learning, 2018. URL https://arxiv.org/ pdf/1802.09477.pdf.
> > > > > >
> > > > > > [4] Mguni David, Jafferjee Taher, Wang Jianhong, Perez-Nieves Nicolas, Yang Yaodong, Yang Tianpei, Taylor Matthew, Song Wenbin, Tong Feifei, Chen Hui and Zhu Jiangcheng and Wang Jun. Learning to Shape Rewards using a Game of Two Partners. AAAI, 2023.

---

### Review · Reviewer_vBun · 2023-03-23

**Summary Of Contributions:**

This paper proposes a new framework, called reinforcement teaching, that casts the problem of learning to learn as a Markov decision process. Specifically, the lower-level learning problem is modeled through a *student*, composed of a parametric function, a learning algorithm, and a measure of performance, while the meta-learning problem is modeled through a *teacher*, which takes actions to improve the sample efficiency of the lower-level problem. The latter structure is enriched by a parametric-behavior embedder, to obtain a compact representation of the student's function parameters, and a learning progress incentive, to shape the rewards of the teacher. Finally, the framework is tested in a curriculum learning problem for reinforcement learning tasks and step-size adaptation for supervised learning tasks.

**Audience:**

Yes

**Broader Impact Concerns:**

This paper can be categorized as fundamental research. I do not think it requires an explicit statement of its potential societal impact.

**Claims And Evidence:**

No

**Requested Changes:**

1) Can the authors address the comments above?

2) Can the authors explicitly specify in the manuscript what are the actual novel contributions of the work? Especially, it is not clear if the idea of framing the learning to learn problem as an MDP is a fully novel contribution, as there seem to be previous attempts in the same direction (see a tentative list below), albeit not as general perhaps.
- Zoph and Le. Neural architecture search with reinforcement learning, 2017
- Zhu et al. Gradient descent optimization by reinforcement learning, 2019
- Jomaa et al. Hyp-rl: Hyperparameter optimization by reinforcement learning, 2019
- Biedenkapp et al. Dynamic algorithm configuration: foundation of a new meta-algorithmic framework, 2020
- Sabbioni et al. Meta Learning the Step Size in Policy Gradient Methods, 2021

**Strengths And Weaknesses:**

**Strengths**

- The proposed framework is general enough to include several learning to learn instances;
- In principle, the proposed framework allows to transfer any advancement in reinforcement learning to the meta-learning problem, albeit requiring a non-trivial design of the MDP in practice;
- The paper reports experiments and results with a careful assessment of their statistical significance, as well as several ablations of the algorithm's components.

**Weaknesses**

- Although I am not familiar with the meta-learning literature, the notion of framing learning to learn as an MDP does not appear to be fully novel, and the paper could be doing more in highlighting actual novel contributions;
- Whereas unifying various meta-learning problem into a unique framework can provide clear benefits, especially for the analysis of problems and methodologies, pursuing a unique algorithmic routine for every problem is not necessarily a good idea, e.g., some instances might be solved more efficiently exploiting their specific structure;
- The paper sells PE and LP as universal solutions, but it is not guaranteed they will always result helpful/harmless.

The paper is certainly interesting, as it addresses the indisputably relevant problem of learning to learn framing it into a general and flexible framework. However, the idea of providing a universal algorithm for meta-learning might be over-ambitious: The reported results are clearly promising, but it seems optimistic to expect PE and LP to work well in every instance.

**(C1, Hand-crafted design and time-to-threshold)**
Although the paper promises a general framework for modeling learning to learn problems, actually translating an instance into the reinforcement teaching formulation seems to require several non-trivial design choices. Those are not limited to the choice of action space, but also to the reward function itself. Especially, I do not find obvious how to set the $m^*$ threshold in practice, without assuming prior knowledge over the lower-level optimization landscape. Can the authors explain how this difficulty can be overcome?

**(C2, PE does not affect Markovianity?)**
The idea of compressing the student's state in a behavior-dependent representation is reasonable. However, even if the underlying process is Markovian, how can we ensure that the process remains Markovian in the PE states? Moreover, I see several cases in which the process is non-Markovian in the first place: Not only the step-size adaptation with Adam mentioned by the paper, but also curriculum learning in RL, where lower-level algorithms may use replay buffers and critics.

**(C3, LP is a roadblock to exploration?)**
Shaping rewards with LP looks sensible whenever the optimization landscape is convex, i.e., any local improvement in the performance will bring the student closer to the global optimum. However, this is not always the case. Sometimes, going through a region with relatively lower performance is necessary to then reach the global optimum (e.g., exploration in RL). I am wondering whether LP can be harmful in such settings.

---

> ### Author Response · Authors · 2023-04-05
> **Author Response to Reviewer vBun 1/2**
>
> W1: \"the paper could be doing more in highlighting actual novel contributions.\"
> --
>
> This is addressed below, in Requested Change 2 (R2).
>
> W2: \"e.g., some instances might be solved more efficiently exploiting their specific structure; \"
> --
>
> While it is true that certain problems may have structure that
> can be exploited, our experiments suggest that previous work are not
> exploiting any structure. For example, the heuristic state
> representations are extremely coarse grained. In providing a general state
> representation, not only are we able to situate many problems on the same
> abstract state space, we are able to improve over the crude heuristics
> used in problem-specific scenarios.
>
> W3: \"The paper sells PE and LP as universal solutions...\"
> --
>
> While we do not provide a theoretical analysis of
> PE\'s approximation quality, we do know that LP is universal in the sense it is a potential-based
> reward shaping and preserves the optimal policy. Our empirical evidence
> across two problem settings and many experiments suggest that PE is a good inductive bias for learning the policy as
> well.
>
> Requested Changes:
> --
>
> ### R1 \"Comments:\"
>
> 1.  C1: \"Hand-crafted design and time-to-threshold)\"
>
>     There are two important facets that we should separate: setting up
>     the MDP and learning a policy in the MDP.
>
>     A design element is always present in learning a policy. For example, there is
>     rich documentation of design in Atari so that an agent is able to
>     learn a good policy, such as sticky actions and frameskips. This
>     design is separate from a given Atari game which provides an action interface and a goal in the game.
>
>     In setting up the teaching MDP, we are only assuming there is a given
>     meta-learning problem to solve. The meta-learning problem defines how we can change the
>     student (the action) and the desired goal for the student (the
>     reward). The choice of the reward function is not given explicitly,
>     but it is defined implicitly in the meta-learning problem\'s goal.
>     In curriculum learning for RL, the meta-learning problem is getting
>     an agent to solve a hard task quickly by sequencing tasks. The time-to-threshold reward is a faithful representation of this goal.
>
>     For supervised learning, we set up the teaching MDP the same way: the
>     meta-learning algorithm must learn to sequence learning rates to
>     help a student reach a desired performance level. We agree with the
>     reviewer that this is a non-standard meta-learning problem. But this is
>     not a design choice, instead, it is to allow better
>     comparison between the two settings of curriculum learning and
>     step-size adaptation.
>
>     **Additional Experiment** We have since run an additional experiment
>     with a different reward (no threshold $m^*$ or termination) using ADAM as
>     the base optimizer and the hard synthetic classification task. The
>     reward is just the accuracy. Policies with high
>     returns are maximizing the (discounted) sum of accuracy which is
>     approximately the area under the curve. The results,
>     which will be in a revision, depict the same finding: PE provides a
>     state representation that learns a better policy compared to the
>     heuristic state representation and the best constant step-size.
>
> 2.  C2: \"PE and Markovian Property\"
>
>     If $\theta$ is a Markov state representation, then the
>     Parametric-behavior Embedding (PE) of $\theta$ is an approximate
>     Markov state representation. Requiring that $\theta$ is a Markov
>     state representation is described in the paper and holds for SGD
>     without momentum in supervised learning for example.
>
>     Denote the student\'s objective function as
>     $J(\theta) = \mathbb{E}_{x,y \sim p(x,y)} J(f_\theta(x), y)$. We show how PE is approximately Markov
>     by showing that the mini-state (which is the input to PE) can
>     represent $J(\theta)$ (reward) and $\nabla_\theta J(\theta)$
>     (state-transition).
>
>     Note that for any particular $\theta$, the objective function is
>     determined by the input ($x$), output of the student
>     ($f_\theta(x)$), and the target ($y$). Hence, $J(\theta)$ is
>     representable as a function of the mini-state (the input to PE).
>     It remains to show that the objective function after a step of gradient
>     descent is also representable in terms of the mini-state.
>
>     For a linear function, $f_\theta(x) = \theta x$, we have that
>     $\nabla_\theta J(\theta) = \mathbb{E}_{x,y \sim p(x,y)}\left[   \nabla _{f_\theta(x)}J(f_\theta(x), y) x^\top \right]$. This means
>     that $\nabla_\theta J(\theta)$ is also representable in terms of the
>     mini-state, with inputs $x$, outputs $y$, and targets $f_\theta(x)$.
>
>     For deep networks, we require that
>     the mini-state includes the outputs of each layer. However, we have
>     demonstrated empirically that we are able to learn policies using PE from just the
>     student\'s outputs ($f_\theta(x)$) and the targets ($y$). We will add this to 4.2.1

---

> > ### Author Response · Authors · 2023-04-05
> > **Author Response to Reviewer vBun 2/2**
> >
> >
> > ### R1 \"Comments (Continued):\"
> >
> > 3.  C3: \"\[is\] LP is a roadblock to exploration?\"
> >
> >     We know that LP is a potential-based reward shaping and does not
> >     change the optimal policy. But, reward shaping\'s effect on learning
> >     is not fully characterized in the literature. There is some work
> >     suggesting that reward shaping improves sample complexity (Gupta et
> >     al. 2022), but how this connects with exploration is unexplored. Due
> >     to the teacher\'s RL algorithm, which explores with $ε$-greedy,
> >     these local optima will not be present in the asymptotic limit. Like
> >     most RL problems, however, we do not know the optimal policy and
> >     hence we cannot conclude whether we have reached the optimal policy.
> >     All we can claim is that we can learn a good policy quicker with LP
> >     than without LP.
> >
> > ### R2 \"Can the authors explicitly specify in the manuscript what are the actual novel contributions of the work? Especially, it is not clear if the idea of framing the learning to learn problem as an MDP is a fully novel contribution, as there seem to be previous attempts in the same direction (see a tentative list below), albeit not as general perhaps.\"
> >
> > Thank you for the additional references. We recognize that there have
> > some previous works that explore using MDPs and RL for meta-learning.
> > Broadly, our work differs because of its generality and effectiveness
> > across multiple problems and for deep nonlinear networks. Below we have
> > included additional details, and we will include these citations in our
> > submission.
> >
> > -   Zoph and Le. Neural architecture search with reinforcement learning,
> >     2017
> >
> >     Our contribution to meta-learning is distinct from the contribution of
> >     this work. They use an RNN policy to generate a neural network
> >     architecture, the construction of which is modelled as a POMDP with
> >     actions adding layers to the neural network. The states of this
> >     POMDP corresponds to states of an architecture, rather than
> >     parameters evolving in time through a learning rule. REINFORCE is
> >     then used to search over architectures as the loss is not
> >     differentiable with respect to the architecture.
> >
> > -   Zhu et al. Gradient descent optimization by reinforcement learning,
> >     2019
> >
> >     This technical report does not provide enough detail to comment on
> >     how they are using reinforcement learning for step-size adaptation
> >     (neither $\phi$, which determines the state, nor $r$ are defined).
> >     This work, however, is very similar to the seminal work (Li and
> >     Malik 2017b) which, as we describe in our submission, makes use of
> >     the heuristic state representation.
> >
> > -   Jomaa et al. Hyp-rl: Hyperparameter optimization by reinforcement
> >     learning, 2019
> >
> >     As before, this paper makes use of a number of \"meta features\", to
> >     form a heuristic state representation.
> >
> > -   Biedenkapp et al. Dynamic algorithm configuration: foundation of a
> >     new meta-algorithmic framework, 2020
> >
> >     This paper assumes access to the true underlying parameters of the
> >     function approximator (Section 5, Sigmoid) and a reward function
> >     with privileged information in only 2 toy settings. The problem
> >     instances are simple 1d regression problems with only 2 parameters,
> >     making learning relatively easy. Similar ideas have been shown to be
> >     effective in seminal work on learning to optimize (Li and Malik
> >     2017a). As our work and Li\'s followup work have shown (Li and Malik
> >     2017b), these approaches do not scale. This is the motivation for
> >     which we developed the PE state representation.
> >
> > -   Sabbioni et al. Meta Learning the Step Size in Policy Gradient
> >     Methods, 2021
> >
> >     This paper is another instance of recognizing an MDP structure in a
> >     specific problem: that of step-size selection in policy gradient
> >     algorithms. The authors recognize that rewards are generally sparse
> >     and so they propose to use the performance difference between
> >     successive parameter values. This is the same as a potential-based
> >     shaping only when $\gamma = 1$, and hence is similar to learning
> >     progress. The state space of the meta-mdp is correctly identified as
> >     the parameter of the student ($\theta$). However, the experiments
> >     conducted are for linear students only.
> >
> > References
> > ----------
> >
> > Gupta, Abhishek, Aldo Pacchiano, Yuexiang Zhai, Sham Kakade, and Sergey
> > Levine. 2022. "Unpacking Reward Shaping: Understanding the Benefits of
> > Reward Engineering on Sample Complexity." *Advances in Neural
> > Information Processing Systems* 35: 15281--95.
> >
> > Li, Ke, and Jitendra Malik. 2017a. "Learning to Optimize." In
> > *International Conference on Learning Representations*.
> > <https://openreview.net/forum?id=ry4Vrt5gl>.
> >
> > Li, Ke, and Jitendra Malik. 2017b. "Learning to Optimize Neural Nets." *Arxiv Preprint
> > Arxiv:1703.00441*.

---

> > ### Comment · Reviewer_vBun · 2023-04-19
> > **Follow-up**
> >
> > I want to thank the authors for their clarifications. I am reporting below a few follow-up comments.
> >
> > 1) I agree with the authors that any learning problem usually requires a careful, hand-crafted, design. However, what I am pointing out is that this framework doubles down on the design process: We still have to design the lower-level tasks (just like in Atari games) and we *additionally* have to design the MDP of the meta-learning task. Especially, given the generality of the framework, I do not see a clear way to automate the latter design process, such as an algorithmic procedure to construct the upper-level MDP from the meta-learning problem. This might be an option if we focus on specific learning to learn instances instead.
> >
> > 2) From my point of view, saying that PE representations are "approximately Markovian" really means that they are not Markovian. This might not be a critical problem, but I think the paper is doing a lot to show that the meta-learning problem can be casted into an MDP, which is not necessarily true when applying this framework.
> >
> > 3) (R2) Would be correct to say that the main novelty of this paper is not to frame the learning to learn problem as an MDP, but rather to employ PE representations and LP reward shaping in this setting, which was hinted in the literature before?

---

> > > ### Author Response · Authors · 2023-04-20
> > > **Thanks for the follow-up!**
> > >
> > > Thanks for the follow-up:
> > >
> > > 1. We are not sure what you mean by doubling down on the design process. We are explicitly taking a step away from "careful, hand-crafted design" in our work. This is also an advantage of meta-learning. Insetad of hand-engineering hyperparameter schedules, for example, meta-learning partially automates this process through learning. If you mean to say that meta-learning, in general, leads to additional design considerations, then this is indeed true. Our goal is not to do away with design considerations entirely from meta-learning, but Reinforcement Teaching is instead taking a step away from "careful, hand-crafted design" that pervades other RL approaches to meta-learning.
> > >
> > > 2. It is true that "approximately markovian" alone can mean "not-markovian" in the worst case. We do clarify that the PE representation is indeed Markovian for a linear student. An interesting direction for future work would attempt to quantify the approximation error incurred by using the PE representation for non-linear students.
> > >
> > > 4. The drivers of Reinforcement Teaching's empirical success are precisely the PE state representation and the LP reward shaping. Both of which are the primary contriubtions and the focus of our empirical investigations. It would not be entirely correct to say that this was hinted in the literature before. Prior work focused on either the parameter representation or hand-designed heuristic state representations.

---

### Decision · Action_Editors · 2023-05-12

**Recommendation:** Accept with minor revision

**Comment:**

The paper proposes to cast the meta-learning problem as a Markov Decision Process.
The reviewers agree that the idea is interesting and general, and the empirical analysis is well executed.
The reviews raised a series of concerns about the clarity of the presentation, the novelty of the approach, and the lack of motivation for some choices.
The authors put considerable effort into addressing reviewers' concerns and provided a significantly improved version of their paper that solved most of the issues.
So, the paper can be accepted after minor revisions addressing the relationships with related works suggested by the reviewers and highlighting better in the introduction the novel contribution with respect to these works.

**Audience:**

The reviewers agree that this paper is of interest to the TMLR's audience.

**Claims And Evidence:**

The reviewers would like to see more discussion about the pros and cons of the proposed solution, focusing on its limitations and its relations with previous works.